# Structural basis for Epstein–Barr virus host cell tropism mediated by gp42 and gHgL entry glycoproteins

Karthik Sathiyamoorthy[1], Yao Xiong Hu[1], Britta S. Möhl[2], Jia Chen[2], Richard Longnecker[2] & Theodore S. Jardetzky[1]

Herpesvirus entry into host cells is mediated by multiple virally encoded receptor binding and membrane fusion glycoproteins. Despite their importance in host cell tropism and associated disease pathology, the underlying and essential interactions between these viral glycoproteins remain poorly understood. For Epstein–Barr virus (EBV), gHgL/gp42 complexes bind HLA class II to activate membrane fusion with B cells, but gp42 inhibits fusion and entry into epithelial cells. To clarify the mechanism by which gp42 controls the cell specificity of EBV infection, here we determined the structure of gHgL/gp42 complex bound to an anti-gHgL antibody (E1D1). The critical regulator of EBV tropism is the gp42 N-terminal domain, which tethers the HLA-binding domain to gHgL by wrapping around the exterior of three gH domains. Both the gp42 N-terminal domain and E1D1 selectively inhibit epithelial-cell fusion; however, they engage distinct surfaces of gHgL. These observations clarify key determinants of EBV host cell tropism.

[1] Department of Structural Biology, Stanford University School of Medicine, 1201 Welch Road, Stanford, California 94305. [2] Department of Microbiology and Immunology, Feinberg School of Medicine, Northwestern University, Ward 6-Northeast Corner 6-255/254, 303 East Chicago Avenue, Chicago, Illinois 60611. Correspondence and requests for materials should be addressed to T.S.J. (email: tjardetz@stanford.edu).

Herpesviruses are ubiquitous, diverse viral pathogens with a large dsDNA genome encapsulated by nucleocapsid, tegument proteins and a lipid membrane envelope[1,2]. The virion bilayer membrane necessitates membrane merging or fusion before the transfer of the dsDNA viral genome to the host and the onset of infection[3]. For many viruses, such as influenza virus or human immunodeficiency virus (HIV), this membrane fusion and entry process is mediated by one multifunctional envelope glycoprotein that is responsible for both host cell receptor binding and lipid bilayer fusion. In contrast, herpesvirus entry is more complex as these steps in the entry pathway are divided among multiple viral envelope glycoproteins (upwards of three to six)[4,5], which determine receptor specificity, host cell tropism and encode the conserved machinery for driving membrane merger. The herpesvirus entry glycoproteins are mechanistically important for viral entry but also targets of the neutralizing antibody response[6].

The herpesvirus family is divided into three sub-families: alpha-, beta- and gammaherpesvirus; and nine viruses have been identified that infect humans. Epstein–Barr virus (EBV), or Human Herpesvirus 4 (HHV-4), is an important viral pathogen and the prototypical member of the *gammaherpesvirinae* subfamily. EBV is the aetiological agent of acute infectious mononucleosis in children and young adults. EBV is an oncogenic virus, causally associated with several malignancies of immunocompromised individuals (transplant and HIV patients), including lymphoid malignancies such as Burkitt and Hodgkin's lymphoma, and epithelial-cell disorders like nasopharyngeal and gastric carcinomas. These EBV-associated malignancies are representative of its two main physiological target cells, epithelial cells and B cells, where it establishes latency. In addition, EBV is also associated with T/natural killer cell lymphoproliferative disorders manifested as secondary complications in immune-system deficient patients[1,2,7,8].

Efficient EBV entry into B cells involves five different envelope glycoproteins, gp350/220, gp42, gH, gL and gB. Gp350/220 binds to complement receptor 2 (CR2 or CD21)[9] or CD35 (ref. 10), which is not essential for entry but increases the efficiency of virus:cell attachment and entry without activating fusion[2]. gH, gL and gB are considered the 'core' fusion proteins, as they are present in all herpesviruses and are required for membrane fusion and entry[2]. Herpesvirus gHgL is a heterodimeric glycoprotein complex composed of soluble gL and membrane-bound gH with a C-terminal transmembrane domain. gB is the most conserved herpesvirus glycoprotein and it is thought to drive membrane fusion[11]. gB functions as a trimer and belongs to the class III viral fusion protein group[12,13]. Finally, the EBV gp42 protein serves as a viral tropism determinant, promoting the infection of B cells while inhibiting the infection of epithelial cells, through high or low levels on the virion, respectively[14].

gHgL is thought to act as a regulator that triggers gB-mediated fusion after binding to host cell receptors[2,15]. EBV gHgL forms high-affinity complexes with gp42, which activates entry into B cells after engaging host HLA class II receptors, while entry into epithelial cells is thought to be triggered by a direct gHgL interaction with integrin receptors[16]. The gp42 N-terminal domain (residues 33–85) binds gHgL with nanomolar affinity and peptides derived from this domain inhibit epithelial-cell fusion with similar potency[17], suggesting that the gp42 interaction may mask the integrin binding site on gHgL. Crystal structures of the gHgL ectodomain from herpes simplex virus 2 (ref. 18), varicella-zoster virus[19], pseudorabies virus[20] and EBV[21] have been determined. Using single-particle electron microscopy (EM), we have shown that the EBV B-cell entry-triggering complex, consisting of gHgL, gp42 and HLA receptor,

assembles into V/Y shaped 'open' and 'closed' states whose conformation appears important to bring virus-host membranes into closer proximity and trigger membrane fusion[22].

Here, we describe the crystal structure of EBV gH, gL and gp42 bound to an anti-gHgL monoclonal antibody (mAb) E1D1, which selectively inhibits membrane fusion with epithelial cells but not B cells. The structure reveals an extensive binding interface of the extended gp42 N-terminal domain with gH, which tethers its C-terminal, receptor binding domain to the complex. The gp42 C-terminal domain interacts with the gH 'KGD' motif implicated in integrin receptor binding, potentially explaining the ability of gp42 to inhibit epithelial-cell entry. However, the gp42 N-terminal domain, which potently blocks fusion with epithelial cells, does not interact with this 'KGD' motif, implicating additional gHgL regions in epithelial-cell entry. Finally, the E1D1 antibody binds to the tip of the gHgL heterodimer, engaging residues solely in gL that are distinct from gp42 and integrin binding sites. Mutagenesis of E1D1 epitope residues in gL indicates that this region of gHgL also plays a cell-type specific role in epithelial but not B-cell entry. These studies provide insights into the structural determinants of gp42-mediated specificity of EBV infection of host cells and highlight regions of gHgL that are functionally divergent in cell-specific virus entry.

## Results

**E1D1 shows selective inhibition of epithelial-cell fusion.** Previous studies demonstrated that intact E1D1 antibody differentially inhibits EBV-mediated membrane fusion with epithelial cells but has little effect on fusion with B cells[16], suggesting that it may be useful for investigating mechanistic differences in the entry of EBV into these two cell types. We produced the Fab domains of E1D1 by proteolysis (Fig. 1a and Supplementary Fig. 1) and compared the effects of intact E1D1 and E1D1 Fab on membrane fusion with epithelial and B cells (Fig. 1b,c). In cell–cell fusion assays with epithelial cells, intact E1D1 showed a significant inhibition at the lowest concentrations tested, while the E1D1 Fab showed a dose-dependent inhibition (Fig. 1b). The inhibition of fusion is incomplete and converges to a maximum of ~31% at the highest concentration of intact E1D1 (12.5 μg ml$^{-1}$), indicating that E1D1 cannot fully block membrane fusion with epithelial cells, but greatly reduces its efficiency. Intact E1D1 is more effective at inhibiting fusion as compared with the Fab, consistent with the higher avidity of the bivalent antibody. In contrast, cell–cell fusion assays for Daudi B cells carried out in the presence of the intact E1D1 and E1D1 Fab did not show any concentration-dependent effects in fusion (Fig. 1c). Based on these titration data, we used 5 μg ml$^{-1}$ final concentration of intact E1D1 or E1D1 Fab in single-point fusion assays. We confirmed the selective inhibition of EBV-mediated fusion with epithelial cells in three independent experiments (Supplementary Fig. 2). Overall, the host cell specific inhibition of membrane fusion by E1D1 points to underlying differences in the mechanisms of EBV entry into its two cell types, despite the common use of its conserved core fusion glycoproteins gHgL and gB.

**Assembly and crystallization of E1D1Fab/gHgL/gp42 complex.** To determine whether gp42 and E1D1 compete for binding to gHgL, we examined the interaction of E1D1 Fab with gHgL and preformed gHgL/gp42 complexes. E1D1 Fab bound to both proteins as observed by gel filtration chromatography, indicating that the E1D1 epitope does not overlap with the gp42 binding site on gHgL (Fig. 1a, lower panels). The effects of the gp42 N-terminal domain peptide or full-length gp42 on E1D1 binding to gHgL was also examined by surface plasmon resonance (SPR) (Supplementary Fig. 3). The kinetic parameters obtained by a global fit of association and dissociation data obtained with a range

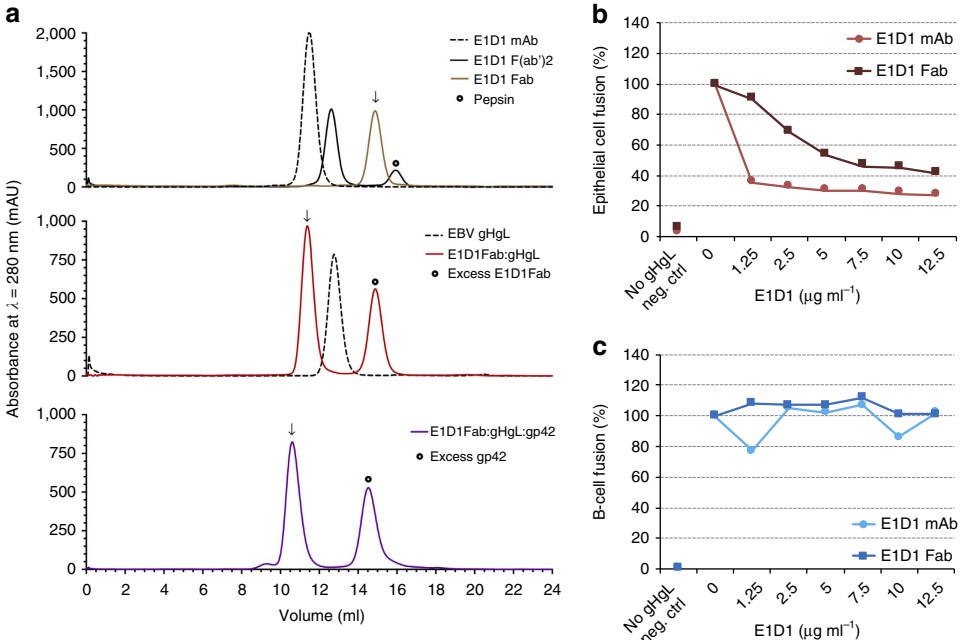

**Figure 1 | E1D1 mAb and Fab bind gHgL and selectively inhibit epithelial-cell fusion.** (**a**) Assembly of E1D1 Fab complexes with gHgL and gHgL/gp42 monitored by size exclusion chromatography. Top panel: E1D1 mAb (dashed black line), E1D1 F(ab')₂ (black line) and E1D1 Fab (brown line). Middle panel: gHgL (dashed black line) and E1D1Fab:gHgL (red line). Lower panel: E1D1Fab/gHgL/gp42 complex (violet line). Arrows mark the major peak fraction of interest. Calibration of Superdex 200 10/300 GL column gave the following elution volume (Ve) versus apparent molecular weight (MW) relationship: $\log_{10}(MW, kDa) = -0.1958*(Ve, ml) + 4.527$ with number of theoretical plates ($N/m$) of 12,137 (nominal value >10,000) and peak symmetry (As) = 1.227 (nominal range, 0.70 < As < 1.30). (**b,c**) Inhibition of fusion activity by E1D1 mAb and Fab. The x-axis indicates the amount of purified E1D1 antibody expressed as a final concentration of E1D1 mAb or Fab (μg ml⁻¹) as present in a luciferase based cell–cell fusion assay using (**b**) epithelial cells (maroon) and (**c**) B cells (blue). No gH and gL plasmids transfected serves as the negative control denoted as neg. ctrl.

---

**Table 1 | Surface plasmon resonance kinetic parameters for binding between gHgL complexes and E1D1 Fab.**

| Stationery phase (ligand) | Surface density of ligand* | Mobile phase (analyte) | $k_a$, M⁻¹·s⁻¹ (×10³) | $k_d$, s⁻¹ (×10⁻³) | $K_D$ (nM) |
|---|---|---|---|---|---|
| gHgL | 1,780 RU† | E1D1 Fab | 18.81 | 2.26 | 120.2 |
| gHgL/gp42 pep‡ | 1,800 RU | E1D1 Fab | 11.32 | 2.41 | 212.9 |
| gHgL/gp42 | 1,110 RU | E1D1 Fab | 26.91 | 2.31 | 85.8 |

gp42 pep has the sequence ⁴⁷KPNVEVWPVDPPPPVNFNKTAEQEYGDKEVKLPHW⁸¹.
*Ligand density rounded off to the nearest tenth after subtracting RU value of baseline level after deactivation step with ethanolamine for the ligand channel minus the corresponding reference channel.
†RU is resonance unit or response unit.
‡gp42 pep is gp42 (47–81) peptide or gp42 35merA1 (ref. 27).

---

of analyte concentrations are collected in Table 1. The results show that the $K_D$ for E1D1 binding was within a factor of 2 for gHgL relative to the gp42 or gp42-peptide complexes, further indicating that E1D1 does not compete with gp42 for binding to gHgL.

Crystallization conditions were identified for the purified E1D1Fab/gHgL/gp42 complex, yielding synchrotron diffraction data to ∼2.9 Å resolution (Table 2). Analysis of the data using the Diffraction Anisotropy Server[23] indicated significant anisotropy along the three crystallographic axes ($a^\star = 2.6$ Å; $b^\star = 3.7$ Å; $c^\star = 2.9$ Å) and the data was therefore truncated and rescaled (Table 2). The structure of the E1D1 complex was solved by molecular replacement (MR) and the E1D1 sequence was determined, allowing model building of the gHgL, gp42 and E1D1 components (Supplementary Figs 4 and 5). The final refinement (Table 2) yielded an overall R-free of 26% with good model geometry and Ramachandran statistics.

**Overview of the E1D1Fab/gHgL/gp42 complex.** The structure of the gHgL, gp42 and E1D1 Fab complex (Fig. 2) reveals key features of the gp42 interaction with gHgL and defines the epitope

recognized by E1D1. The gp42 interaction with gHgL is primarily mediated by the extended gp42 N-terminal domain, consisting of residues 33–85 (Fig. 2a; Supplementary Tables 1 and 2). Electron density for N-terminal domain residues allowed the modelling of gp42 beginning with amino acid 43 (Supplementary Fig. 6). In previous gp42 crystal structures[24,25], this region was disordered suggesting that it is a flexible, unstructured domain in the absence of gHgL. Peptides spanning gp42 residues 33–85 bind gHgL with nanomolar affinity, similar to intact gp42, further indicating that this domain does not fold independently of gHgL. In contrast to prior predictions that the gp42 N-terminal domain might bind to the prominent groove between gH domains D-I and D-II (ref. 21), the crystal structure shows that it adopts an extended conformation wrapping around the exterior of gH domains D-II, D-III and D-IV (Fig. 2b). No gp42 residues engage the D-I/D-II groove (Fig. 2). Since gp42 is a type II membrane protein, its transmembrane domain (residues 7–27) immediately precedes the N-terminal domain. In the gHgL complex, electron density for the gp42 N-terminus is observed at residue 43, which is bound to a subsite in gH D-IV, positioning the transmembrane anchors of

**Table 2 | Data collection and refinement statistics.**

| | E1D1Fab/gHgL/gp42 | |
|---|---|---|
| Beamlines for screening/data collection | Beamline 12-2 SSRL (SLAC) and LS-CAT (APS) | |
| Wavelength, Å (energy, eV) | 0.97872 (12,667.99) at 21-ID-F | |
| Space group | I 2 2 2 | |
| | | |
| *Unit cell dimensions* | | |
| a, b, c (Å) | 105.52, 166.01, 272.41 | |
| α, β, γ (°) | 90, 90, 90 | |
| | | |
| *Data collection statistics (XDS), before (left)* | | |
| *and after (right) anisotropy correction* | | |
| Resolution, (Å) | 46.11–2.90 (3.005–2.901) | 46.11–3.10 (3.211–3.100) |
| Total reflections | 311,400 (30,732) | 255,673 (25,751) |
| Unique reflections | 53,195 (5,222) | 39,682 (4,336) |
| Redundancy | 5.9 (5.9) | 5.8 (5.9) |
| Completeness, % | 99.83 (99.05) | 90.61 (61.37) |
| I/sI | 12.21 (1.60) | 14.44 (2.88) |
| Wilson B-factor | 77.97 | 51.84 |
| R-merge | 0.1232 (1.59) | 0.1032 (0.852) |
| CC ½ | 0.997 (0.586) | 0.998 (0.811) |
| CC* | 0.999 (0.86) | 0.999 (0.946) |
| Ellipsoid truncation resolution limits | NA | $a^* = 2.6$ Å, $b^* = 3.7$ Å, $c^* = 2.9$ Å |
| | | |
| *Refinement statistics (Phenix 1.10-2155)* | | |
| Resolution (Å) | | 46.11–3.10 (3.211–3.100) |
| Reflections used | | 39,681 (2,663) |
| Reflections used for R-free | | 1,977 (128) |
| $R_{work}/R_{free}$ | | 0.2332/0.2681 (0.3246/0.3598) |
| Number of atoms | | |
| Total | | 10,618 |
| Protein | | 10,562 |
| Ligand/ion | | 56 |
| Water | | 0 |
| B-factors | | 55.67 |
| Protein | | 55.34 |
| Ligand/ion | | 119.21 |
| Water | | NA |
| Ramachandran statistics | | |
| Total accepted, % | | 98.7 |
| Outliers, % | | 1.3 |
| r.m.s. deviations | | |
| Bond lengths (Å) | | 0.005 |
| Bond angles (°) | | 0.83 |
| Molprobity clashscore | | 8.74 |
| Number of TLS groups | | 34 |
| Rotamer outliers (%) | | 9.9 |

APS, advanced photon source; CC, cross-correlation; CCD, charge-coupled device; eV, electron-volt; LS-CAT, Life Sciences-Collaborative Access Team; NA, not applicable; r.m.s, root mean square; SLAC, Stanford Linear Accelerator Center; SSRL, Stanford Synchrotron Radiation Lightsource; TLS, Translation-Libration-Screw-rotation model.
Data collected from a single crystal. Values in parenthesis are for the last resolution shell.
$R = \Sigma ||Fobs|-|Fcalc||/\Sigma|Fcalc|$, where R-work is calculated from this general formula using all reflections included in the refinement of the crystal structure model, and R-free is calculated from a five per cent random sample of reflections not included at any stage in the refinement of the crystal structure model.

gp42 and gH at the same end of the complex, consistent with their orientation relative to the viral membrane.

The E1D1 Fab binds at the tip of gHgL D-I, aligned with the longest axis of the gHgL dimer (Fig. 2). Previous mutagenesis studies demonstrated reduced binding of E1D1 to gH mutants L65A and L69A (ref. 26), placing the expected epitope close to the gHgL D-I/D-II interface and the 'KGD' motif. However, gH resides L65 and L69 do not contact E1D1, indicating that their effects on E1D1 binding are indirect and likely due to perturbation of the gHgL D-I structure. Furthermore, the E1D1 Fab only contacts residues in gL, indicating that its specificity for both gH and gL is due to its recognition of a tertiary epitope formed by the co-folding of gL with the N-terminal residues of gH. The binding of E1D1 at the tip of the gHgL heterodimer is distant from the 'KGD', motif and the putative site of integrin receptor binding (Fig. 2), suggesting that E1D1 inhibition of epithelial-cell entry does not involve a direct competition with the gHgL interaction with integrins.

The structures of gp42 and gHgL observed here are similar to previously determined X-ray structures[21,24,25]. Superposition of the gp42 C-type lectin domain (CTLD) with the previously determined structures of gp42 alone (3FD4)[24] or in complex with HLA-DR1 (1KG0)[25] yields an overall root mean square deviation (r.m.s.d.) of 0.7 Å between Cα carbons (Supplementary Fig. 7a). The largest structural difference occurs in the gp42 '171 loop', where E171 forms a salt bridge interaction with the lysine in the gH 'KGD', motif. This loop is lower in gp42 alone (3FD4) and lowest in gp42/HLA-DR1 (1KG0). In addition, the gp42 hydrophobic pocket (HP) is in its widest state in our current structure, due to movements in the '188 loop' and '206 loop'

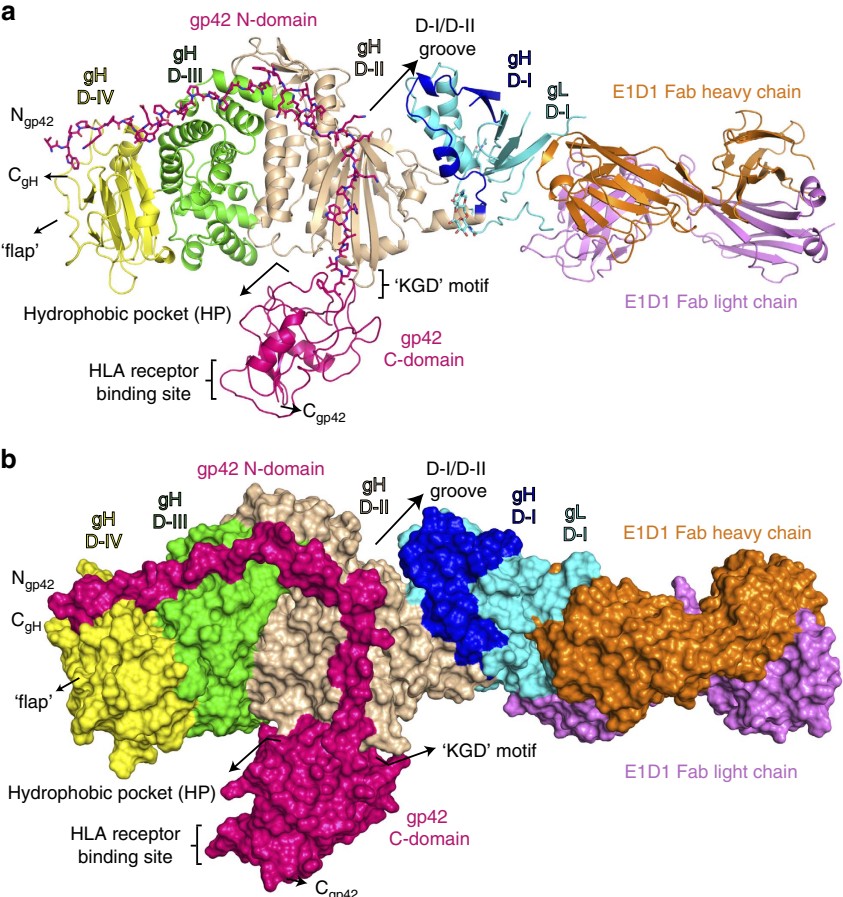

**Figure 2 | Crystal structure of EBV gHgL/gp42 bound to the E1D1 Fab.** The structure is shown in (**a**) cartoon and (**b**) surface representations coloured as follows: gL (cyan) is part of domain I (D-I), gH D-I (blue), D-II (wheat), D-III (green), D-IV (yellow); gp42 (hotpink), E1D1 Fab heavy chain (orange) and E1D1 Fab light chain (violet). Structures were rendered using MacPyMol.

(Supplementary Fig. 7a). The gp42 '158 loop' adopts a conformation similar to the HLA-DR1 bound state.

The superposition of gHgL with our previous model of EBV gHgL alone (3PHF)[21] similarly aligns closely with an overall r.m.s.d. of only 1.5 Å between gH Cα carbons (Supplementary Fig. 7b). The largest deviation is in the helical tilts of gH D-II helix 2α-6 and gH D-III helix 3α-9. These shifts in gH could result from the gp42 HP and N-terminal domain interactions with gH, causing these gH domains to shift relative to each other and inducing the helical tilts. However, the relatively low resolution of the gHgL structures (3.58 Å for 3PHF and 3.1 Å here) may also contribute to some of these observed differences.

**gp42 N-domain interactions with gH.** The high-affinity anchoring of gp42 to gHgL is critical for the activation of membrane fusion with B cells, as mutations in the gp42 N-terminal domain that weaken this interaction lead to a reduction in fusion activity. Both gp42 and gp42-derived peptides potently inhibit membrane fusion with epithelial cells with nanomolar affinity[14,17,27]. The overall gH interaction with gp42 buries a total of 2,600 Å$^2$ (compared with 2,170 Å$^2$ for gH:gL) of which 1,400 Å$^2$ is buried only by the gp42 N-terminal domain (Fig. 2).

We previously mapped gp42 N-terminal domain residues critical for the high-affinity binding with gHgL[27], dividing the domain into two segments separated by a linker region of 5 amino acids (residues 62–66). Alanine scanning mutagenesis

identified pairs of gp42 residues that are critical for high-affinity binding[27]. Consistent with these functional studies, we observe that the gp42 N-terminal domain interaction can be subdivided into 5 High-Affinity Binding Determinants (HABDs) on gH, referred to as HABD-1 through 5 (Fig. 3). HABD-1 through HABD-3 engage the N-terminal half of the gp42 N-terminal domain, while HABD-4 and HABD-5 interact with gp42 residues after the linker (62–66; Fig. 3a). HABD-1 lies within gH D-IV, HABD-2 at the junction of gH D-III and D-IV and HABD-3 lies within D-III. HABD-4 lies at the junction of gH D-II and D-III domains, while HABD-5 is formed by residues within gH D-II.

The HABD-1 site engages gp42:W44 and gp42:P46 through van der Waals interactions and a single main-chain hydrogen bond (Fig. 3b). gp42:W44 nestles into a hydrophobic cavity surrounded by gH D-IV residues gH:I602, gH:F605, gH:L660, gH:I613 and gH:F614. gH:I602 and gH:F605 are part of the D-IV 'flap' region (Fig. 2) while gH:I613 and gH:F614 are held firmly in place in a short loop containing the conserved disulfide bridge C612-C615. gH:L660 is within the D-IV β9 strand (referred to as 4β-9). gp42:P46 is positioned directly above gH:I613 without making significant contacts to any other gH residues. This D-IV region has been shown to be important in membrane fusion and it has been hypothesized that the D-IV flap may undergo a conformational change to promote fusion[20]. However, no conformational changes in the D-IV flap have been observed yet and the binding of the gp42 residues to the HABD-1 site would be expected to further stabilize this region.

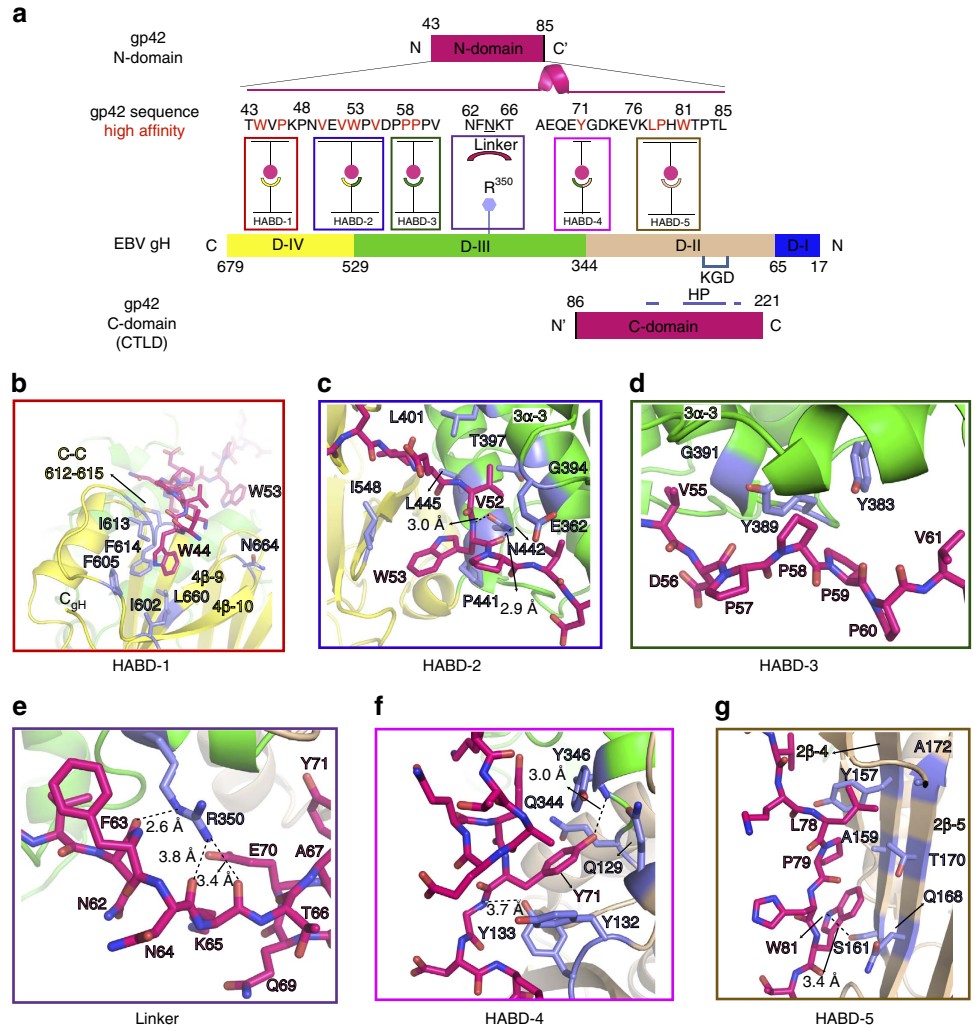

**Figure 3 | Interactions of the gp42 N-terminal domain with gH.** (**a**) Schematic for the gp42:gH interaction. The gp42 N-domain (43–85) and C-domain (86–221) are shown above and below gH domains, positioning them adjacent to gH domains with which they interact. Residues in gH interacting with the gp42 N-domain[27] are boxed and labelled as HABDs 1–5. The most important gp42 N-domain residues are in coloured red and in bold type. Underlined residue gp42:N64 is potentially N-linked glycosylated. gH:R350 lies underneath the gp42 linker (residues 62–66) and is shown as a hexagon. The putative integrin binding motif in gH ('KGD' motif) is shown in light blue. (**b–g**) Each of the primary high-affinity binding subsites (HABDs) of gH for the gp42 N-domain peptide is shown separately, highlighting the interacting residues. (**b**) HABD-1, (**c**) HABD-2, (**d**) HABD-3, (**e**) Linker, (**f**) HABD-4 and (**g**) HABD-5. Structures were rendered using MacPyMol.

The HABD-2 binding interaction is the most extensive, with both side chain and main chain atoms forming hydrogen bonds and van der Waals contacts between gp42 and gH (Fig. 3c). gp42:V50, gp42:V52, gp42:W53 and gp42:V55 make direct interactions to gH. The flanking gp42:V50 and gp42:V55 residue interactions with gH, along with gp42:P54, may help position this peptide segment. The gp42:V50 amide makes a backbone hydrogen bond to the carbonyl from gH:T635. gp42:V55 approaches the 3α-3 gH helix closely at gH:G391 and gH:G394, assisted by a hydrogen bond between the gp42:V55 amide with side chain oxygen of gH:E362. gp42:V52 is central to this binding interaction within HABD-2. The methyl groups of gH:T397, gH:L401 and gH:L445 on one side and gH:G394 on other side lock the gp42:V52 side chain in place, while the Cβ group of gH:E362 buttresses the interaction from above. The main chain amide and carbonyl groups of gp42:W53 make bi-dentate hydrogen bonds with the gH:N442 side chain, while the indole ring of gp42:W53 makes lateral non-polar interactions with gH:P441 and gH:I548.

The HABD-3 site features two tyrosines, gH:Y383 and gH:Y389, in the loop between gH helices 3α-2 and 3α-3, interacting with the gp42 poly-proline motif, gp42:P57 to

gp42:P60 (Fig. 3a,d). gp42:P57 and gp42:P60 flank the interaction site, facing outwards, while gp42:P58 and gp42:P59 form contacts with the gH tyrosines in a manner reminiscent of poly-proline interactions with SH3 domains[28].

The HABD-1–3 sites are followed by a linker region of five gp42 amino acids ([62]NFNKT[66]). The length, but not the sequence, of the linker were shown to be important for high-affinity binding[27]. The structure reveals that the gp42 linker is hoisted above the gH surface, in part due to gH:R350, which is the gH residue with one of the most buried accessible surface area (as calculated by Naccess[29]) resulting from gp42 binding (Fig. 3e and Supplementary Table 1). gH:R350 makes multiple hydrogen bonds to the gp42 main chain, consistent with the lack of side-chain specificity for the linker region. gp42:N64 is a N-linked glycosylation site, although no density for carbohydrate moieties was observed, and it is positioned above gH:R350 with the asparagine side chain projecting away from the gH surface (Fig. 3e). Immediately following the linker is a short α-helix ([67]AEQE[70]), the only secondary structural feature of the N-terminal domain. This helical structure may account for the residual binding of gp42 linker deletion mutants[27], as the

helix could potentially unfold, extending the N-terminal domain region to allow interactions with the flanking HABD sites.

HABD-4 is at the interface of gH D-II and D-III and primarily involves the sequestering of gp42:Y71 (Fig. 3f) by a cluster of three gH tyrosines (gH:Y132, gH:Y133 and gH:Y346) and gH:Q344. The hydrophobic portions of the gH residues form a

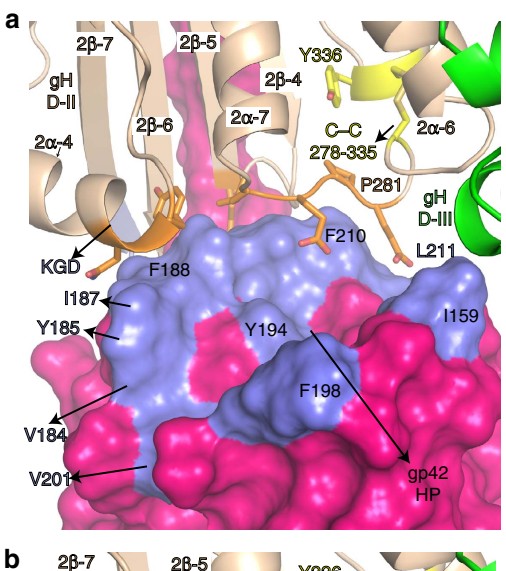

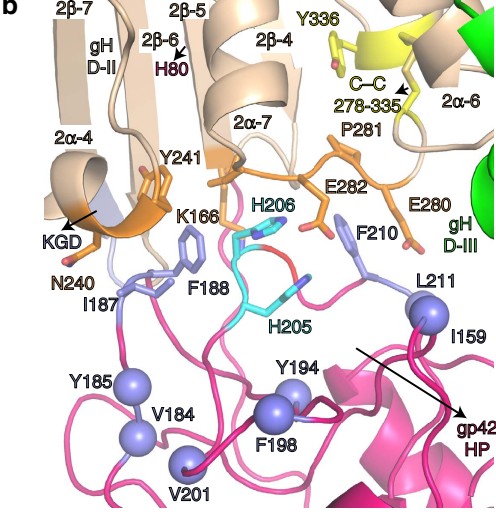

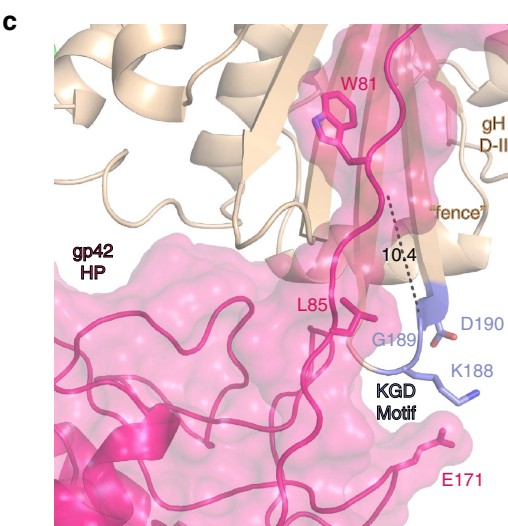

well-defined pocket and make van der Waals contacts with the side chain of gp42:Y71. At the bottom of the pocket, the gp42:Y71 hydroxyl group forms a hydrogen bond to backbone carbonyl of gH:Q344.

HABD-5 is primarily formed by residues in two antiparallel gH β strands in D-II (2β-4 and 2β-5; Fig. 3g), capped at one end by gH:D165 in the β-turn connecting the strands and at the opposite end by gp42:T136 and gp42:M137 in an extended chain. The gp42 chain travels parallel to the two β strands, with the side chains of gp42:L78, gp42:P79 and gp42:W81 facing the gH β-sheet and forming hydrophobic contacts with gH residues. gp42:L78 forms packing interactions with gH:Y157, gH:T170 and gH:A172 side chains and gp42:P79 with gH:A159, while gp42:W81 also makes polar contacts to the gH:S161 side chain through its indole nitrogen. Immediately before gp42 forms the interactions with HABD-5, the gp42 peptide chain makes a sharp, perpendicular turn (Fig. 2), guided by salt bridge interactions of gp42:E70 and gp42:E75 with gH:R350 and gH:R152, respectively. gH:I134 also makes a significant contribution, holding this turn through backbone hydrogen bonds to gp42:K74 and gp42:V76.

**gp42 C-domain interactions with gH**. The gp42 C-terminal domain (86–221) adopts a CTLD fold that binds HLA class II receptor, as well as forming interactions with gH (Fig. 4a). Negative stain single-particle EM studies of the gHgL/gp42/HLA entry triggering complex[22] indicated that the gp42 CTLD could form direct interactions with gH through a HP that had previously been postulated to consist of residues gp42:I159, gp42:V184, gp42:Y185, gp42:I187, gp42:F188, gp42:Y194, gp42:F198, gp42:V201, gp42:F210 and gp42:L211 (Fig. 4a,b). The observed gp42 CTLD interface with gH is formed by only one edge of this predicted gp42 HP, making direct contacts with gH through 3 of these 10 HP residues (gp42:I187, gp42:F188 and gp42:F210; Fig. 4a,b). Residues gp42:I159, gp42:Y185, gp42:V184, gp42:V201, gp42:F198, gp42:Y194 and gp42:L211 do not form contacts with gH (Fig. 4b). The gp42 C-terminal domain buries a total of 1,200 Å² of gH surface area, but does not mediate a high-affinity interaction. The gp42 HP makes contacts with four loops within gH D-II, between helices 2α-4 and 2α-5, helices 2α-6 and 2α-7, β strands 2β-4 and 2β-5 and β strands 2β-6 and 2β-7 (Fig. 4a,b). gH:N240 and gH:Y241 from D-II, at the end of the 2α-4 helix, interact with the Cβ atoms of gp42 residues gp42:I187 and gp42:F188. The gp42:F210 side chain, which has been shown by mutagenesis to be important for membrane fusion[30], inserts into the centre of the loop between helices 2α-6 and 2α-7, against the cavity formed by an unusual central loop proline (gH:P281, Fig. 4a,b).

Residues at the junction of the gp42 N- and C-terminal domains interact with the [188]KGD[190] motif in gH D-II (exposed loop between 2β-6 and 2β-7), which is the putative integrin

**Figure 4 | The gp42 HP interacts with gH D-II and overlaps the integrin binding 'KGD' motif.** (**a**) One edge of the previously identified gp42 HP interacts with gH D-II. The gp42 C-domain surface is coloured hotpink, with HP residues shown in slate (shade of blue) and labelled. The gH D-II is shown as a cartoon and coloured wheat. Gp42 residues gp42:I187, gp42:F188 and gp42:F210 interact with gH D-II. (**b**) Two gp42 histidines (gp42:H205 and gp42:H206, cyan sticks) in the HP form contacts with gH:E282 (orange). HP residues not directly in contact with gH are indicated by their Cα atom shown as spheres. (**c**) gp42:L85 (sticks) contacts gH:G189 in the [188]KGD[190] motif, while gp42:E171 (cyan) is positioned to form charge–charge interactions with gH:K188. The shortest distance (10.4 Å) between the minimal inhibitory peptide of gp42 peptide ending at gp42:W81 (ref. 17) with the gH [188]KGD[190] motif is highlighted. Structures were rendered using MacPyMol.

receptor-binding site required for epithelial-cell entry (Fig. 4c). The residues connecting the gp42 N- and C-terminal domains (residues 84–88) extend along one face of the 'KGD' motif, and along with gp42:E171 of the CTLD domain, could interfere with integrin binding. However, these gp42 interactions with the 'KGD' motif fail to explain the ability of shorter gp42-derived peptides, truncated to residue 81, to inhibit membrane fusion with epithelial cells[17,31]. Residue gp42:W81 is >10 Å away from the 'KGD' motif (Fig. 4c), suggesting that these gp42-derived peptides do not inhibit membrane fusion by blocking integrin receptor binding. The gp42 N-terminal domain might inhibit integrin binding allosterically, by affecting the conformation or flexibility across gH D-II to D-IV that could be important to membrane fusion, or it may block a secondary interaction site necessary for triggering fusion. In support of the latter possibility, gp42:E75 forms a salt bridge interaction with gH:R152 at the edge of the D-I/D-II groove (Supplementary Table 1) and the mutation of gH:R152A has been shown to selectively reduce fusion with epithelial cells[32].

**E1D1 exclusively engages residues in gL.** The E1D1 Fab contacts residues in gL but not gH (Fig. 5), although E1D1 binding is dependent on the co-expression and folding of both gH and gL[33]. Residues L65 and L69 in gH, which disrupt E1D1 binding when mutated to alanine[26], do not contact E1D1, but form extensive hydrophobic interactions with gL residues immediately underlying the E1D1 epitope (Fig. 5). Mutations in these gH residues likely disrupt the proper folding of D-I.

The E1D1 complementary determining regions (CDRs) form a cradle that engages gHgL, with H-CDR1, H-CDR2 and L-CDR1 rising up to form the sides and H-CDR3, L-CDR2 and L-CDR3 defining the cradle base (Fig. 5). Segments from both the gL N-terminus and C-terminus dock into this cradle. The C-terminal gL residues 127–135 lie against L-CDR1–3, with the gL:W133 side chain tucked into a pocket at the base of the cradle. gL:W133 is sandwiched by $V_H$:Y100 in H-CDR3 on one side and $V_L$:Y37 in L-CDR1 and $V_L$:K55 in L-CDR2 on the other side (Fig. 5 and Supplementary Fig. 8a). gL:R135 forms salt bridge interactions with $V_H$:D31 in H-CDR1, while $V_L$:N35 and $V_L$:Y37 in L-CDR1

form hydrogen bond interactions to the carbonyls of gL:N129 and gL:A132 (Supplementary Fig. 8a). The N-terminal residues of gL (27–33) are located within the cradle but make few contacts to E1D1.

An additional interaction is formed external to the central cradle (Fig. 5 and Supplementary Fig. 8b), where H-CDR1 forms a parallel β-strand with the gL β-3 strand (residues 72–79). The side chain of E1D1 $V_H$:S25 acts as a hydrogen bond donor to the main chain of gL:N72, while the main chain of H-CDR3 residues $V_H$:G26 and $V_H$:T28 make hydrogen bonds to gL:L74, gL:V75 and gL:S77 in Lβ-3. The side chains of gL:L74 and gL:I76 pack into a shallow hydrophobic patch to the side of H-CDR1 (Fig. 5 and Supplementary Fig. 8b).

**Mutations in gL affect fusion with epithelial cells.** Because the E1D1 epitope is distant from the putative integrin receptor-binding site, we hypothesized that partial inhibition of epithelial-cell fusion by E1D1 (Fig. 1b,c) might be due to an unanticipated functional role of this D-I region. To test this possibility we mutated residues in D-I within and adjacent to the E1D1 epitope and tested their effects on gHgL expression, membrane fusion and recognition by a panel of conformation-specific antibodies (Fig. 6).

The gL mutants generally exhibited selective effects in epithelial-cell fusion and only minor effects on B-cell fusion (Fig. 6a). All of the mutants were expressed similar to wt gHgL as measured by western blotting (Fig. 6b and Supplementary Fig. 9) and at the cell surface, as measured by an HA-tag present on gL (Fig. 6a). Mutations in a loop neighbouring the E1D1 interface, which contains a glycosylation site at residue gL:N69 showed significant effects with epithelial-cell fusion alone (Fig. 6a). Mutations that removed glycosylation at residue gL:N69 (gL:N69L, gL:S71V or gL:N69L/S71V) selectively increased fusion with epithelial cells 2–3 fold, while membrane fusion with B cells was unaffected (Fig. 6a). However, the gL:N69D mutation, which also blocks glycosylation (Fig. 6b), did not affect epithelial-cell fusion (Fig. 6a). While removal of the glycosylation at gL:N69 increases fusion activity, incorporation of a negative charge at residue 69 counteracts this effect. Mutations gL:L74E, gL:R78M and gL:Y131A selectively

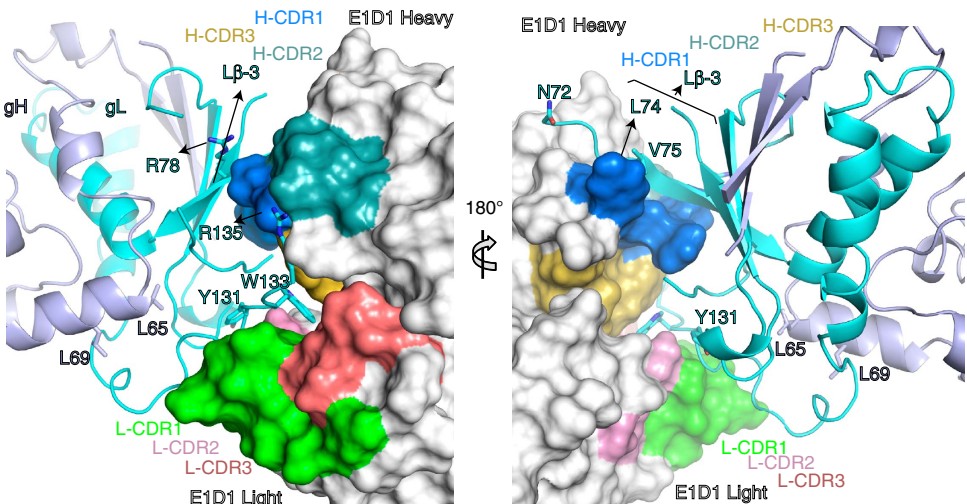

**Figure 5 | E1D1 binds exclusively to residues in gL.** E1D1 interacts with N- and C-terminal gL residues through a cradle formed by E1D1 CDR regions. Certain key gL residues (cyan) are shown in stick representation. The gH residues gH:L65 and gH:L69 in the D-I/D-II linker helix that affect E1D1 binding[26] are also shown. The right panel is a 180°rotation from the left view as indicated. The gL β-3 strand (Lβ-3, residues 72–79) is shown forming contacts along the exterior of the E1D1 H-CDR1 in surface. gL:L74-I76 is buried in a shallow hydrophobic cleft within H-CDR1. E1D1 is shown as a surface (white) with CDRs coloured as indicated here: H-CDR1 is coloured marine, H-CDR2 is light teal, H-CDR3 is sulfur; and, L-CDR1 is green, L-CDR2 is pink, and L-CDR3 is deep salmon (colouring scheme label as from PyMol). H-CDR2 and L-CDR3 are not visible in the right figure panel. gH is coloured in light blue and gL in cyan. Structures were rendered using MacPyMol.

reduced fusion with epithelial cells to ~50% of wild-type (wt) levels (Fig. 6a), while B-cell fusion remained near wt levels. The gL:L74A, gL:I76A and gL:R78A mutations individually reduced fusion to 70%, while the triple mutant (gL:L74A/I76A/R78A) reduced fusion levels to 40% of wt. The gL:Y131A mutation

reduced epithelial fusion levels to ~50% while gL:Y131F maintained wt fusion activity, indicating the importance of an aromatic group at this position. The gL:R78L and gL:I76D mutants were the only gL mutations that reduced both B-cell and epithelial-cell fusion comparably (Fig. 6a).

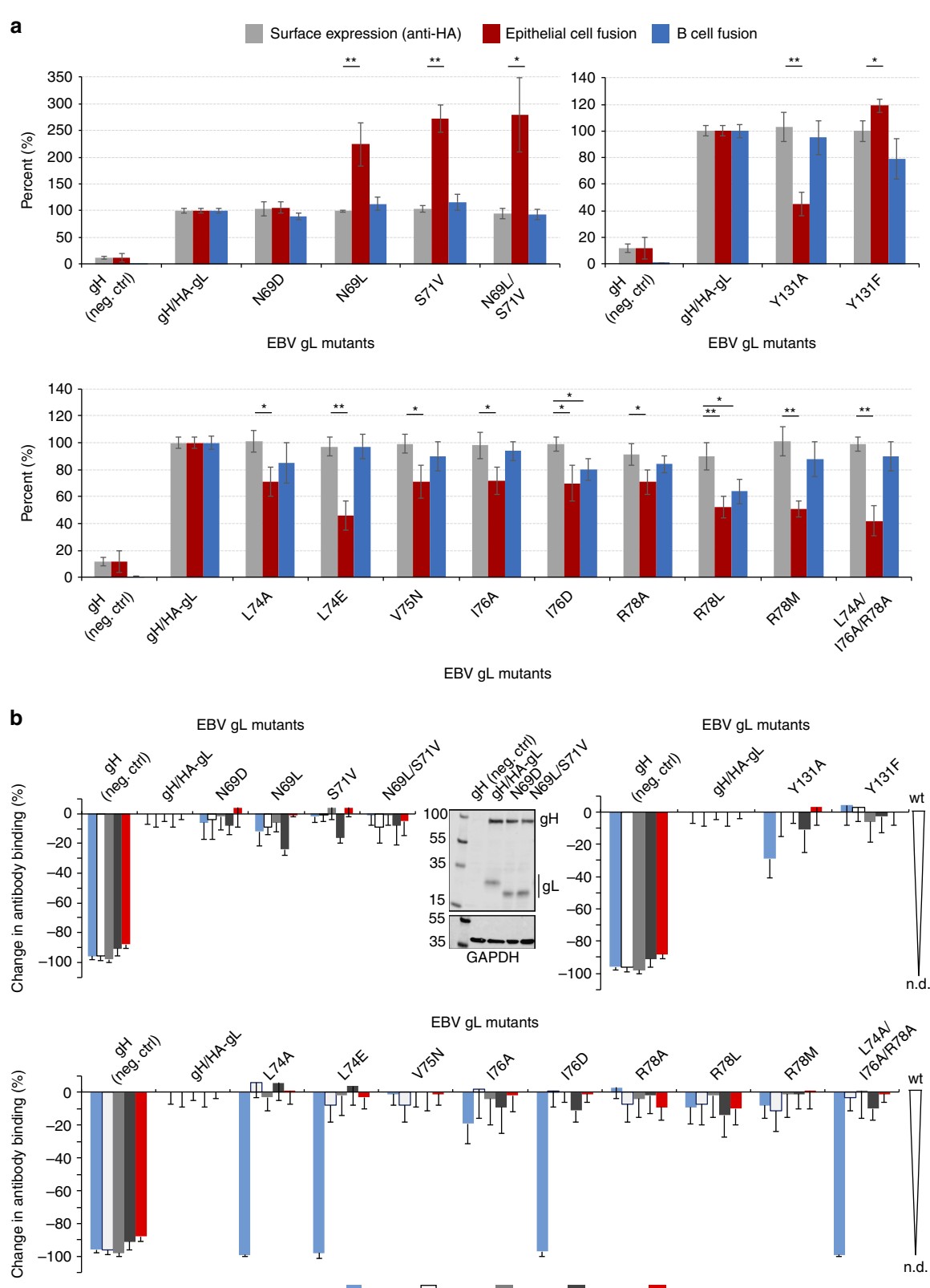

The conformational state and surface expression of the gHgL mutants were also examined using three conformation-sensitive monoclonal antibodies (E1D1, CL40 and CL59) and a polyclonal anti-gHgL serum (HL800). All of the mutants behaved similarly to wt gHgL, except for four mutants containing substitutions of residues lying within the E1D1 epitope (Fig. 6b). The gL mutants gL:L74A, gL:L74E, gL:I76D and gL:L74A/I76A/R78A were all defective in E1D1 binding, but not in binding to other anti-gHgL antibodies CL40, CL59 and HL800 indicating that the overall folding and surface expression of the mutants was similar to wt (Fig. 6b). gL residues 74 and 76 interact with a hydrophobic surface on H-CDR1 (Supplementary Fig. 8b) and substitution of gL:L74 to alanine or glutamic acid reduced E1D1 binding. For residue gL:I76, only the gL:I76D mutant disrupted E1D1 binding (Fig. 6b and Supplementary Fig. 8b).

Overall, the data demonstrate that gL residues within the E1D1 epitope and in this neighbouring loop predominantly show cell-specific effects on membrane fusion with epithelial cells, which is similar in magnitude to the selective inhibition observed with E1D1. This region of D-I at the tip of gHgL may therefore play a role in epithelial cell entry that enhances, but is not absolutely required for membrane fusion. However, these mutational data do not exclude the possibility that E1D1 may independently affect gHgL interactions with integrin receptors and thereby also inhibit membrane fusion.

## Discussion

The structure of the gHgL/gp42 complex bound to the E1D1 Fab reveals insights into interactions and structural regions that regulate EBV entry into epithelial and B cells, defining the molecular basis for EBV tropism (Fig. 7). The structure reveals that the gp42 N-terminal domain does not bind to the D-I/D-II groove, as previously hypothesized[21]. Instead the gp42 N-terminal domain adopts an extended conformation and wraps around the external surfaces of three of the four gH domains, tethering the receptor binding C-terminal domain to the complex. Peptides derived from the N-terminal domain, which are potent inhibitors of EBV entry into epithelial cells, do not physically obstruct the putative integrin receptor 'KGD' motif, although the intact gp42 does. The gp42 C-terminal domain interacts with gH near this 'KGD' motif through its exposed HP[24,25], as anticipated by low-resolution EM studies[22], but only 3 of the 10 residues lining this pocket contact gH directly. Finally, the E1D1 antibody, which partially inhibits membrane fusion with epithelial cells, but not B cells[16], binds at the tip of gHgL D-I, distant from the integrin-binding motif in D-II. Mutations in this D-I surface have similar selective effects on epithelial-cell fusion as E1D1, suggesting that E1D1 engages a site in gHgL that can enhance, but is not absolutely required for, epithelial-cell fusion.

Gp42 is a modular adaptor that binds gHgL with high affinity, blocking entry into epithelial cells, while promoting EBV entry into B cells. The ability of gp42 to activate EBV fusion with B cells depends on three distinct interactions of gp42: (1) high-affinity binding to gHgL, mediated by the extended N-terminal domain; (2) weak interactions between the gp42 HP and gH; and (3) high-affinity binding to HLA class II molecules on B cells (Fig. 7a). The high-affinity interaction of gp42 with gHgL is achieved by docking the gp42 N-terminal domain into five distinct subsites distributed over three domains of gH. Disruption of these interactions blocks the ability of gp42:HLA complexes to stimulate membrane fusion, indicating that the tethering of gHgL to these complexes is required. However, mutations within the gp42 HP interaction site with gH can also block the activation of membrane fusion. These mutations do not affect the high-affinity gp42:gHgL or gp42:HLA interactions, demonstrating that gHgL and gp42 proximity is required but not sufficient to promote membrane fusion after HLA binding.

Our previous low-resolution EM studies of the entry triggering complex (gHgL/gp42/HLA) indicated that the complex could exist in 'open' and 'closed' states, pivoting around an interaction between the gp42 HP and gH[22]. In the 'closed' state, the gHgL and gp42/HLA molecules adopt an acute orientation, with their C-terminal ends co-localized to one side of the complex (Fig. 7b). In the 'open' state, representing ∼50% of the complexes, the angular orientation of the HLA relative to gHgL is highly variable with the longest axes of the gHgL and gp42/HLA components adopting larger angles in some cases >90° (Fig. 7b)[22]. Studies of gp42 HP mutants suggested that the 'closed' state may be important to activating membrane fusion. Thus the role of the gp42 HP interface to gH may be to stabilize a conformation which positions the transmembrane domains of gH and HLA into sufficiently close proximity to promote membrane fusion mediated by the gB protein. The crystal structure reported here aligns more closely with the 'closed' conformational state of the complex (Fig. 7a,b). We note that one edge of the gp42 HP engages gH, with two gp42 histidines (gp42:H205 and gp42:H206) that are poised to interact with gH:E282 (Fig. 4b). These residues could contribute a pH-dependent salt bridge interaction under the low pH of endosomes, which could further stabilize an activated state for the receptor-bound gHgL/gp42 complexes to promote membrane fusion.

Both intact gp42 and gp42-derived peptides block membrane fusion with epithelial cells with nanomolar potency[17,31], thereby regulating the tropism of EBV. For entry into epithelial cells, integrin receptors are thought to directly engage a 'KGD' motif exposed in a loop in gHgL D-II. Docking of the structure of the αvβ6 integrin onto gHgL through this motif results in a plausible model for the epithelial-cell receptor complex that shows similarity to gHgL/gp42/HLA 'open' conformations (Fig. 7c). We anticipated that the interactions of gp42 with gHgL might block access of integrin to this 'KGD' motif, which could explain the ability of gp42 to regulate EBV tropism. Although intact gp42 could obstruct the access of integrin receptors on epithelial cells to a 'KGD' motif in gH (Figs 4c and 7c); shorter, inhibitory gp42-derived peptides do not contact or obscure this motif.

**Figure 6 | Mutations in the E1D1 epitope affect membrane fusion with epithelial cells.** (a) Cell–cell fusion assays with gL mutants. The gH vector in the absence of gL was used as a negative control (denoted as neg. ctrl). Surface expression of the mutants was monitored using HA-tagged gL (grey bars). Fusion activity is expressed as a percentage of wt activity levels for epithelial-cell fusion (maroon coloured bars) and B-cell fusion (blue coloured bars). The y-axis represents a shared scale for both surface expression levels (%) and fusion activity (%). Multiple t-tests, one per each row (see Supplementary Table 3), was performed with statistical significance determined by the Holm–Sidak method with α = 0.05. Each row is analysed individually, without assuming a consistent s.d. P value style: <0.05 (*), <0.01 (**), <0.001 (***), <0.0001 (****). Error bars are ± s.d. (b) gHgL mutant surface expression and conformation was monitored using anti-gHgL E1D1, CL40 and CL59 monoclonal antibodies and HL800 polyclonal serum using a CELISA assay. The results are expressed as % change in antibody binding relative to wt, with most antibodies showing no changes in binding to the mutants relative to the wt gHgL (values near zero represent wt binding). E1D1 binding is selectively lost for mutations in gL residues gL:L74, gL:I76 and gL:Y131. n.d., not detected. Error bars are ± s.d. A western blot of the gL N69 glycosylation mutants detected using a polyclonal anti-gHgL antibody (rabbit) is shown adjacent to the antibody binding data, with monoclonal anti-GAPDH blots shown as a loading control. The gel shows the expected shift in molecular weight of the gL mutants due to loss of glycan at the mutated NXS motif.

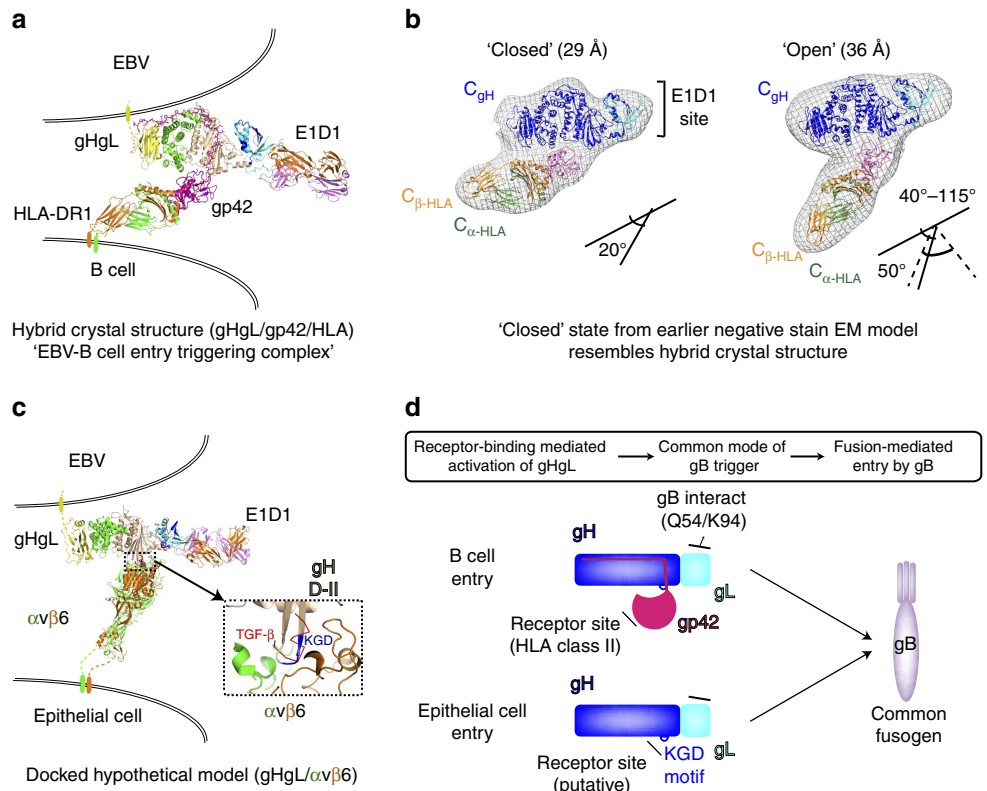

**Figure 7 | Structure based models for initiating stages in EBV entry and host receptor dependent cell tropism.** (**a**) Structural models of the EBV B-cell entry-triggering complex. A hybrid crystal structure generated by aligning the gp42 C-domain, observed in the crystal structure described here, with the gp42:HLA-DR1 complex (1KG0)[25]. (**b**) This composite of crystal structures closely mirrors the 'closed' state of gHgL/gp42/HLA complex observed by previous negative stain single-particle EM[22]. This result and a variation of this figure appeared in a previous publication[22]. (**c**) A hypothetical model for the epithelial-cell entry complex, based on docking the gH 'KGD' motif onto the 'RGD' motif in the αvβ6 crystal structure with TGF-β (4UM9)[49]. (**d**) Schematic of the host-cell tropism mediated by gHgL complexes converging on the activation of gB-mediated membrane fusion.

These observations suggest that the inhibitory action of gp42 involves either an allosteric inhibition of integrin binding or, alternatively, the inhibition of a secondary interaction at a site that is distinct from the 'KGD' motif.

The E1D1 antibody selectively blocks EBV fusion with epithelial cells, but not B cells, although this inhibition is partial and saturates at ∼70% inhibition. The crystal structure demonstrates that the E1D1 epitope does not overlap the gH 'KGD' motif implicated in epithelial-cell entry and models of the complexes of E1D1 with integrin-bound gHgL (Fig. 7c) do not show any steric conflicts between the antibody and integrin receptor. However, this simple docking model may fail to capture structural details that could account for the partial E1D1 inhibition. We identified mutations of gL residues within the E1D1 epitope region that also partially block epithelial-cell fusion and observed that the removal of a neighbouring carbohydrate site in gL enhances epithelial, but not B-cell fusion. These mutational data point to gHgL D-I as having a cell-specific function in epithelial-cell entry that can enhance membrane fusion. It remains to be established how this D-I site mediates these cell-specific effects, but D-I could for example promote viral glycoprotein interactions, or potentially interact with an unknown co-receptor on epithelial cells, to enhance fusion. Interestingly, in human cytomegalovirus (HCMV) gHgL, this N-terminal region is the site for assembly of gO and UL128–131 proteins, which regulate HCMV infection of fibroblasts, epithelial and endothelial cells[34]. The assembly of the EBV tropism factor, gp42, onto gHgL is clearly distinct from these HCMV complexes, with the gp42 receptor binding domain positioned in the middle of the gH structure and its tethering domain extended through to the

C-terminus of gH. Nonetheless, the N-terminal domains of both HCMV and EBV gHgL complexes appear to retain some functional characteristics in cell-specific entry that remain to be fully understood.

Neither gp42 nor E1D1 interact with gL residues Q54 and K94 that have been implicated in homotypic interactions and activation of gB[35]. These residues are left unhindered in the E1D1Fab/gHgL/gp42 complex and available for activating B-cell entry (Fig. 7d). Presumably this gHgL-gB activation step is conserved for epithelial-cell entry as well. Although we favour a model in which gB activation is a convergent and conserved step in both B-cell and epithelial-cell entry (Fig. 7d), it remains possible that distinct interactions between gHgL and gB could be required to maintain efficient epithelial-cell versus B-cell entry. The observed differences in fusion activity of gL mutants in this study with B cells versus epithelial cells could potentially represent mechanistic differences in the requirements for gHgL-gB interactions and activation. Finally, the high-affinity binding pockets (HABD-1 through 5) in gH for the gp42 N-domain that we have identified here may provide targeted sites for the development of small molecule and peptidomimetic inhibitors to block EBV infection, reactivation and amplification in infected individuals and transplant recipients.

## Methods

**Protein expression and purification for complex formation.** Details for the construct design, expression and purification of soluble ectodomain of EBV gHgL heterodimer, and gp42 with a N-terminal six histidine tag have been previously published[22,31] but described briefly here. EBV gH residues 18–679 and gL residues

22–137 were transferred to pBacgus4x-1, with gH and gL under the expression control of p10 and polyhedron promoters respectively in opposing directions. Cloning introduced three residues AMT for gH and AMD for gL at their respective N-termini resulting in a final 10,713 bp pBacgus4x-1 vector. Sf+ (Protein Sciences Corp), and High Five (Thermo Fisher) cells were maintained at a density of 1.5, and 1.0 million cells per ml with HyClone media (Sf+) and ESF921 media (High Five) every 2 days shaking at 135 r.p.m. and maintained at 27 °C. To infect cells at a density of 1.8 million cells per ml, 2% v/v final of gHgL baculovirus stock to Sf+ cells or gp42 baculovirus stock to High Five cells was added. The cells were left shaking and media supernatant collected by centrifugation after 72 h (3 days).

Gp42 was purified from the supernatant by immobilized metal affinity chromatography (IMAC) using $Ni^{2+}$ or $Co^{2+}$ (Talon) based IMAC resins that bind to the six histidine tag. Before IMAC, media supernatant was pH adjusted with 50 mM final Tris pH 8.0, and precipitated with 5 mM final calcium chloride and 1 mM final nickel or cobalt chloride. A second centrifugation step removed the precipitate before the supernatant was passed through the IMAC resin. Bound gp42 was eluted from the IMAC column with up to 300 mM imidazole in a 50 mM Tris, 300 mM NaCl pH 8.0 containing buffer.

The untagged gHgL protein was purified from supernatant with an E1D1 antibody affinity column as described previously[22,31]. E1D1 mAb expressing hybridoma cells were a generous gift from Lindsey Hutt-Fletcher. The hybridoma was expanded to produce E1D1 mAb in the supernatant using the National Cell Culture Center (Biovest/NCCC). E1D1 mAb was then purified from the clarified supernatant by protein G resin (L00209, Genscript), followed by gel filtration with Superdex 200 (GE Life Sciences). Purified E1D1 mAb was conjugated to ultralink hydrazide matrix resin following manufacturer's instructions (Product 53149, Pierce, Thermo Scientific). 1X phosphate buffered saline (PBS) pH 7.4 (Corning) was used for washing and 1X PBS pH 7.4 with 0.02% (w/v) sodium azide used for storing the E1D1 mAb column. Bound gHgL was eluted with gentle Ag/Ab elution buffer pH 6.6 (Pierce/Thermo Scientific) and immediately buffer-exchanged into 1X PBS pH 7.4 for injection into Superdex 200 10/300 GL gel filtration column (GE Life Sciences). Both gp42 and gHgL soluble proteins were purified in the final buffer 20 mM Tris, 150 mM NaCl, pH 7.4 before crystallization. Schematics of the proteins are shown in Supplementary Fig. 4.

For protein used in SPR binding kinetics assays, 1X PBS pH 7.4 was used as the gel filtration buffer to avoid interference by a primary amine based buffer in the ligand immobilization step following the amine-coupling method. Protein yields were 700–900 µg l$^{-1}$ of cells for gHgL and >3 mg l$^{-1}$ of cells for gp42 after the final gel filtration step. Excess gp42 was snap frozen in liquid nitrogen for long-term storage to avoid proteolytic degradation of the gp42 N-terminal domain. For crystallization trials, gp42 was either used directly or cleaved with enterokinase enzyme (EKMax, Thermo Fisher) to remove the six histidine tag. Successful crystallization hits for the complex were obtained here with the his-tagged gp42.

### Enzymatic fragmentation of E1D1 mAb.
For crystallographic studies, the E1D1 (IgG2a subclass) mAb was enzymatically digested to generate the Fab fragment for a monovalent complex with gHgL/gp42. Both papain (from papaya latex, P3125-25MG Sigma) and pepsin (from porcine gastric mucosa, P6887-1G from Sigma) were used to generate the E1D1 Fab fragment. Ficin, while commonly used for antibody fragmentation, did not yield the expected 50 kDa Fab fragment, likely due to its specificity for the IgG1 subclass. Papain digestion of E1D1 mAb was carried out at 1:5 w/w ratio (excess E1D1) with antibody buffer-exchanged into 0.1 M sodium citrate pH 6.0, 10 mM EDTA, 10 mM cysteine.HCl (freshly made) for the digestion step. Fab fragments were generated with an overnight digestion (16 h) at 37 °C, followed by separation of the undigested, Fab and Fc fragments by protein A resin (in 1X PBS pH 7.4) and gel filtration chromatography. Pepsin digestion was carried out at a 1:40 w/w ratio (excess E1D1) in a low-pH buffer consisting of 100 mM sodium acetate pH 4.0 at 37 °C for 16 h. Digestion resulted in divalent F(ab')2 fragment that needed to be subsequently reduced to monomers. Reduction and alkylation to generate stable Fab' fragments from pepsin digestion was carried out in two different treatments. First, by using 5 mM Tris(2-carboxyethyl)phosphine.hydrochloride pH 7.0 (TCEP.HCl, M02624 from Oakwood Chemical) reduction on ice for up to 1 h, followed by 50 mM final iodoacetamide (IAA, I6125-5G from Sigma) for 30 min–1 h. Second, by using a mild reducing agent, 2-mercaptoethylamine (2-MEA, Product 20408 Pierce, Thermo Scientific) at 50 mM final concentration in PBS–EDTA as reaction buffer (manufacturer instruction), split into multiple eppendorf tubes each with ~500 µg (500 µl volume) E1D1 antibody incubated at 37 °C for 6 h. In both cases, reduction was followed by gel filtration to purify the monomeric Fab' fragment. E1D1 Fab' generated from pepsin digestion and reduced/alkylated with TCEP/IAA treatment gave larger crystals of the complex with subsequent optimization for the high-resolution synchrotron data. Although the E1D1 fragment used in the structure determination was the monomeric Fab' fragment generated by pepsin, instead of Fab, no electron density for the extra hinge residues were observed. The papain-generated Fab fragment also gave crystals under identical conditions. For simplicity, we refer to the E1D1 fragment as the Fab in the text.

### Crystallization of E1D1Fab/gHgL/gp42 glycoprotein complex.
Individual proteins (E1D1 Fab, gHgL, gp42) were purified in large scale from cell culture supernatants as described above, concentrated and the final complex formed by gel filtration, with an elution volume (Ve) of 10.6 ml (Superdex 200/GE Life Sciences).

The final protein concentration was ≈7.5 mg ml$^{-1}$ (A$_{280 nm}$, Nanodrop) in final buffer 20 mM Tris, 150 mM NaCl, pH 7.4. All crystallization trials were done at room temperature (22 °C) using various sparse matrix crystallization screens (Hampton and Qiagen) with the Phoenix robot (Art Robbins Instruments). Optimized crystals grew in 0.1 M Tris pH 8.0, 0.18 M potassium citrate, 10–16% (w/v) PEG 3350 with and without 3–15% (v/v) PEG 600. Crystals appeared within the third day and grew larger over the course of 2 weeks. Slender rod-shaped crystals were carefully looped and frozen in ice-free liquid nitrogen after a quick soak in original mother liquor with 15% (v/v) PEG 600. Initial crystals all diffracted at best to only about 4 Å. Crystal annealing, room temperature diffraction and dehydration of crystals did not improve diffraction. Diffraction resolution was improved when the crystals were grown in PEG 6000 instead of PEG 3350 as the precipitant. Synchrotron X-ray diffraction data was still anisotropic, but a complete dataset was collected with reflections visible beyond 3 Å to enable the structure solution and model building.

### Structure solution and refinement of E1D1Fab/gHgL/gp42 complex.
The crystals belong to spacegroup I222 with 1 complex per asymmetric unit. Data reduction was done using X-ray Detector Software (XDS)[36]. Matthew number (Vm)[37] calculation within ccp4 software[38] gave a value of 3.23 with 62% solvent content (0.99 probability) for an estimated approximate molecular weight of 185 kDa. MR using the program Phaser[39] was successful with a single-gHgL heterodimer (3PHF), gp42 CTLD residues 95–221 (1KG0, chain C) and a murine Fab (1PLG, IgG2a, kappa light chain with an elbow angle of 190°) as search models used together in a single run. Different Fab models with varying elbow angles were selected and prepared by Sculptor[40] to match the target (E1D1) Fab sequence and to truncate flexible regions (CDR loops) for MR. The Fab elbow angle made a stark difference between success and failure of obtaining the structure solution. The log-likelihood gain (LLG) measure from Phaser was positive and increased successively with each component placed and the translation z-scores were all >12.0 both indicative of a successful structure solution. Electron density for the gp42 N-terminal domain and E1D1 CDR loops was apparent in Fo − Fc difference electron-density maps. Phenix autobuild[41] was used to perform iterative model building, refinement and density modification, leading to an improved electron density map. Missing residues were then manually built in Coot using C-alpha baton mode and adding terminal residue in an iterative fashion[42]. Phenix refine was used for structure refinement to minimize positional (xyz) parameters and individual temperature factors (B factors) between data and model[43,44]. To improve the refinement, anisotropic scaling with ellipsoid truncation[23] was carried out on the unmerged intensities output from XDS. The resultant data had cutoffs in resolution of $a^* = 2.6$ Å, $b^* = 3.7$ Å, $c^* = 2.9$ Å. Examples of the final electron-density maps for building gp42 and E1D1 are shown in Supplementary Fig. 6. All relevant data collection and refinement statistics are collected in Table 2. The refinement was performed with a resolution cutoff of 3.1 Å in light of the completeness in the last resolution shell. The final R-work and R-free for the structure are 0.23 and 0.26 respectively. Feature-enhanced map (FEM) strictly followed the 2mFo-DFc map as expected and accentuated side chain density especially at the molecule surface helping in rotamer fits and small backbone corrections in early interpretations[45]. Alternate side-chain conformation for gH:H426 in gH D-III core is distinct and can be modelled attesting to the quality of the electron density map. N-linked glycosylation sites to build two N-acetyl glucosamine (NAG) sugar units each can be seen at gH residue N60 and gL residue N53, projecting from opposing sides of D-I in the rod shaped molecule. Crystallography programs used here were installed and updated through the SBGrid consortium portal[46].

### SPR-binding kinetics.
Binding kinetics assay to determine on-rate (k$_a$), off-rate (k$_d$) and affinity (K$_D$) between gHgL or gHgL/gp42 N-domain peptide or gHgL/gp42 and E1D1 Fab was performed using a 404pi biosensor instrument (BiOptix, CO). 1X PBS pH 7.4 with 0.05% (v/v) Tween-20 was used as the running buffer. EBV glycoproteins (as the 'Ligand') were immobilized onto a carboxy-methyl dextrose (CMD-200 m; BiOptix, CO) biosensor chip by amine-coupling method. Sensorgrams with different serial dilution (1:3), that is, concentration series of E1D1 Fab as the mobile analyte were flown over the ligand and the sensorgram data fit globally to a 1:1 interaction model using GraphPad Prism 7. Kinetic parameters from the model fit are collected in Table 1. Sensorgram traces with the model fit overlaid on the data are shown in Supplementary Fig. 3.

### Cell-based membrane fusion assays.
Virus free cell–cell fusion assays were performed as previously described[47]. Wt protein-fusion levels (positive control) in each experiment were set to 100% and the effect of gL mutants or added antibody compared. CHO-K1 effector cells (ATCC CCL-61 or CRL-9618) were transfected with plasmids for luciferase reporter under T7 promoter control and either gB and gH, gL for epithelial-cell fusion or gB, gH, gL and gp42 for B-cell fusion. Twenty-four hours post transfection with Lipofectamine 2000 (Invitrogen), the cells were washed, detached, counted and mixed 1:1 with target cells stably expressing T7 RNA polymerase as well as intact E1D1 mAb or E1D1 Fab were added. Target cells were either Daudi-T7–29 cells (ATCC CCL-213) to mimic B-cell fusion or HEK293-T14 cells (ATCC CRL-3216) for epithelial-cell fusion, which stably

express T7 RNA polymerase. The mixed cells were cultured in 24-well plates in Ham's F12 media with 10% heat inactivated fetal bovine serum (FBS). After 24 h, the cells were washed with PBS and lysed with 100 µl of passive lysis buffer (Promega). Luciferase activity or luminescence was quantified from 20 µl of lysed cells with 100 µl of luciferase assay reagent (Promega) in a 96-well plate on a Perkin-Elmer Victor plate reader. Experiments were carried out in triplicate (biological replicates) or simple for the E1D1 titration curves. The gHgL western blots were developed with a rabbit polyclonal anti-gHgL antibody generated by immunizing rabbit with pSG5-EBV-gH and -gL plasmids and subsequently boosting with purified gHgL protein. The GAPDH western blots were developed using a commercial anti-GAPDH antibody (Abcam ab8245; 1:5,000 dilution).

**Cell enzyme-linked immunosorbent assay (CELISA).** The cell-surface expression of gHgL mutants was determined by CELISA as described in previous reports[48]. CHO-K1 cells were transfected with different gL mutants and gH wt plasmid. Twenty-four hours post transfection, $4 \times 10^4$ cells per well were transferred to a 96-well plate and incubated for another 24 h. The cell-surface expression of gHgL was evaluated using conformation specific anti-gHgL antibodies: E1D1, CL40 and CL59 monoclonal antibodies and HL800 polyclonal serum (kindly provided by Lindsey Hutt-Fletcher, used as 1:500 dilution). After incubation with primary antibody for 30 min and fixation with 2% formaldehyde and 0.2% glutaraldehyde in PBS for 15 min, an anti-mouse or anti rabbit biotin-labelled secondary antibody (Sigma) was added at 1:500 dilution and incubated for 30 min. After washing, streptavidin-labelled horseradish peroxidase (1:20,000) was further incubated with the fixed cells for 30 min. Peroxidase substrate, 3,3′,5,5′-tetramethylbenzidine (TMB) one component HRP Microwell Substrate, TMBW-0100-01 (BioFX, Surmodics) was added and the amount of cell surface staining was determined by measurement at 380 nm with Perkin-Elmer Victor plate reader.

**SDS-PAGE migration assay.** CHO-K1 cells were transfected with control, or the plasmids as indicated in figure legends (Fig. 6b and Supplementary Fig. 9). After 24 h, transfected cells in 6-well plates were collected and lysed in 200 µl Lysis Buffer (20 mM Tris-HCl, pH 7.4, 100 mM NaCl, 1 mM EDTA, 5 mM MgCl₂, 1% Triton X-100 and Calbiochem's 1 × protease inhibitor cocktail set I). The cell lysates were clarified by centrifugation and 100 µl of the lysates were mixed with 100 µl 2X SDS loading buffer (60 mM Tris-Cl pH 6.8, 0.2% SDS, 25% glycerol, 0.01% bromophenol blue). The samples were loaded onto a BioRad 4–20% mini PROTEAN TGX gel for western blotting. After electrophoresis, proteins were transferred to nitrocellulose membranes (Schleicher & Schuell, Keene, NH). The blots were blocked with 5% nonfat dry milk in TBS buffer (20 mM Tris-HCl, pH 7.6, 137 mM NaCl) for 2 h at room temperature. The blots were washed with TBS and incubated with primary antibodies anti-gHgL (a polyclonal antibody serum generated by immunizing rabbit with pSG5-EBV-gH and -gL plasmids and subsequently boosting with purified gHgL with QED Bioscience Inc., 1:100 dilution) or anti GAPDH (Abcam clone 6C5, 1:5,000 dilution) overnight at 4 °C. Anti-rabbit IRDye800 or anti-mouse IRDye680 secondary antibodies (LI-COR biosciences, Lincoln, NE) were added to the membranes at a dilution ratio of 1:10,000 and incubation was continued for 1 h at room temperature. Protein bands on the membrane were visualized with the Odyssey Fc Western blotting imager using Image studio version 2.0 (LI-COR biosciences, Lincoln, NE).

**Data availability.** The E1D1Fab/gHgL/gp42 co-ordinates and structure factors have been deposited in the Protein Data Bank (PDB) under the accession code 5T1D. The DNA sequences of the E1D1 antibody have been submitted to Genbank under accession codes KX755644 (E1D1 heavy chain) and KX755645 (E1D1 light chain).

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

## Acknowledgements

We thank the members of the Jardetzky and Longnecker laboratories for their help and support. This research was supported by AI119480 and AI076183 (R.L. and T.J.) from the National Institute of Allergy and Infectious Diseases, and by CA117794 (R.L. and T.J.) from the National Cancer Institute. We thank Lindsey Hutt-Fletcher for providing the E1D1, CL40 and CL59 hybridomas. This research used resources of the Advanced Photon Source, a U.S. Department of Energy (DOE) Office of Science User Facility operated for the DOE Office of Science by Argonne National Laboratory under Contract No DE-AC02-06CH11357. Use of the LS-CAT Sector 21 was supported by the Michigan Economic Development Corporation and the Michigan Technology Tri-Corridor (Grant 085P1000817). Use of the Stanford Synchrotron Radiation Light-source, SLAC National Accelerator Laboratory, is supported by the U.S. Department of Energy, Office of Science, Office of Basic Energy Sciences under Contract No DE-AC02-76SF00515. The SSRL Structural Molecular Biology Program is supported by the DOE Office of Biological and Environmental Research, and by the National Institutes of Health, National Institute of General Medical Sciences (including P41GM103393). The contents of this publication are solely the responsibility of the authors and do not necessarily represent the official views of NIGMS or NIH.

## Author contributions

K.S., Y.X.H., B.S.M., J.C., R.L. and T.S.J. conceived and designed the experiments; K.S., Y.X.H., B.S.M. and J.C. performed the experiments; K.S., Y.X.H., B.S.M., J.C., R.L. and T.S.J. analysed the data; K.S., Y.X.H., B.S.M. and J.C. contributed reagents/materials/analysis tools; K.S., Y.X.H., B.S.M., J.C., R.L. and T.S.J. wrote and edited the manuscript.

## Additional information

**Competing financial interests:** The authors declare no competing financial interests.

