## [Peer Review File · Nature Communications]

Reviewers' comments:

Reviewer #1 (Remarks to the Author):

Review of Sathiyamoorthy et al. 2016

This is an highly important paper that clarifies multiple EBV gHgL hypotheses promoted in previous works by these and other authors. At the moment this reviewer believes that the manuscript needs significant reorganization and clarification. And as a non-crystallographer I would like to see fewer structures, a recognition that many are repeated, and most importantly to bring the conclusions out of the supplement and directly into the body of the paper. This manuscript is incremental using the term in its most favorable commendable light..the authors have pursued this area and have brought it to a most satisfying "three-dimensional" conclusion. This is a major contribution! Comments that I believe will help conveying the importance of their results follow.

The authors need to put all of these new concepts in one place, i.e., a table which summarizes the findings. Now they are spread though-out the manuscript. Major findings and questions to be addressed;

1) gp42 N-terminal domain does NOT bind gH DI/DII groove as previously thought. In fact, no gp42 residues engage this groove.

2) Only 3 of the previously 10 suggested gp42 residues from the hydrophobic pocket contact gH

3) gp42 peptides that block membrane fusion with epithelial cells do NOT physically block the KGD motif, as was previously assumed. The authors suggest that these gp42 peptides indirectly affect gHgL-integrin binding. Has it actually been shown that KGD is where integrin binds to gH? Or is it just assumed? In an earlier paper the authors mutated EBV gH KGD to AAA but they only tested gp42 binding, not integrin binding. Was this tested here and if not why?

4) Antibody E1D1 does NOT bind near the KGD motif (gH residues L65 and L69) as previously

thought. E1D1 does not contact these residues at all. Their effect on E1D1 is therefore indirect (due to structural changes in the gH DI domain). In fact, E1D1 binds only gL residues.

Furthermore, E1D1 does NOT block integrin binding directly based on the crystal structure (because it does not occlude KGD). However, E1D1 binding has been shown to reduce (but not eliminate) integrin binding. Authors suggest this is another structural effect (it's also known that E1D1 binding changes the conformation of gHgL (Borza et al. 2011)).

5) The gH C-terminal "flap" motif is covered when gp42 binds. Does this change the authors' thinking in the possible function of the gH flap? It had been suggested that this region opens to reveal hydrophobic residues, that then interact with the membrane to promote fusion. If gp42 is covering the flap, it can't open.

6) Implications...Does your crystal structure (gHgL/gp42/E1D1) differ from either previously published structures (gHgL alone or gp42 alone)? E1D1 has been previously shown to alter gHgL conformation, so are there any significant changes? This needs to be addressed.

7) "Closed" vs "open" conformational state of gHgL/gp42. The authors claim this crystal is of closed state (line 235). Please remind the reader what this means... does "closed" or "open" align with a gHgL function?

8) Clearly, all of the action presumably occurs on one side of gH/gL but what about the other side of the complex? Are there changes that take place in structure when gp42 is bound?

1) To repeat...Supplemental Fig. 6 is too important to be in the supplement. It needs to be part of the main paper. It summarizes many of the authors' findings.

2) They do a great job at correcting published papers, including their own:

3) gp42 does not bind a groove in gH between DI/DII (the crystal struct paper)

4) the epitope for the E1D1 Ab was suggested to involve residues L65 and L69 (Hutt-Fletcher) but in the structure the epitope does not involve these residues. the effect is more indirect, due to perturbations in the gHgL structure.

5) But I would like to see more correlations between this structure, the EM structure and the predictions they made previously. And these should be in the Discussion, not in the Results section.

6) Most figure legends are very wordy and repeat the information in the text. Because of this, rather than being informative they become tedious.

7) Fig1b, c: what does the x-axis mean? Is that total amount? ug/ml? ug/ul?

8) Why not show parts of Suppl Fig2 in Fig1?

9) Fig 2C is redundant. Information conveyed could be shown in Fig 2B.

10) Description for Fig3 is hard to follow, full of details and too long

11) Fig6 gave me the most trouble, may be because this was the only section that I could actually follow. The legend is wordy and confusing.

12) What is the point of it? So, mutations in the E1D1-gHgL interface were made and these mutants have decreased binding to the Ab. The assumption is that these mutants will also have an effect in function. Why?

13) The statement that "the data demonstrate cell-specific modulation" is a bit strong. The effect of these mutations is modest. The data are supposed to add functional weight to the structure but fail to do that.

14) I don't understand what Fig6A is: ELISA? SPR? It looks like ELISA, but it is graphed in such an odd way that I do not know if the negative bars mean no binding or just decreased binding.

15) The assumption that by western all mutants have wt expression is only partially correct. As they are going to asses them for function, I would have liked to see surface levels by ELISA in both epithelial and Daudi cells. What was the Ab used for detection?

- 16) What is the control in fusion? It is not indicated in the legend, text or methods.
- 17) The description for Fig6c first covers residues 74-131 and then the glycosylation sites. Therefore, the graphs should be reversed as well.
- 18) This figure feels like a filler, to have some functional data that was inspired by the structure. Not a very impressive end to the paper.
- 19) I could not find any reference in the text for Suppl Fig 3 and 4.
- 20) Nevertheless, Suppl Fig 4a is a repeat of Fig 2a.
- 21) I found Suppl Fig6 more informative than some of the structural descriptions. As it is described and discussed at length, it should be part of the main set of figures.

Reviewer #2 (Remarks to the Author):

The manuscript NCOMMS-16-11713-T reports the crystal structure of the EBV gH/gL/gp42 glycoprotein complex bound to the Fab of an anti-gH/gL antibody E1D1. The manuscript thus focuses on two sets of interactions: one between gH/gL and gp42, and the other between gH/gL and E1D1 Fab. Although the gH/gL/gp42 complex, which is necessary for entry into B cells, has been characterized previously, this is its first atomic-level structure. The structure reveals how gp42 binds gH/gL: the N-terminal extension of gp42 wraps around gH, stably tethering the C-terminal domain that engages HLA receptor on B cells. This is an unexpected, and exciting finding. Furthermore, the structure shows that the integrin-binding KGD motif on the surface of gH is partly occluded by gp42, which lends support to the hypothesis that gp42 blocks EBV entry into epithelial cells by competing with integrin receptors for binding to gH/gL.

The structure of gH/gL/E1D1 complex is interesting because E1D1 inhibits entry into (or fusion with) epithelial cells but not B cells. The structure shows that despite being specific for gH/gL complex, E1D1 Fab makes direct contacts only to gL but not gH, which suggests that gH is required for proper folding of the E1D1 epitope. It is less clear why E1D1 inhibits entry into epithelial cells. The E1D1 binding site, located at the tip of the D-I domain of gH/gL, is not in the vicinity of the putative integrin-binding site, and the authors hypothesize that the D-I domain

"may form cell-type specific interactions that enhance membrane fusion with epithelial cells". Besides certain integrins, no other molecules on the surface of the epithelial cells have been proposed to be involved in EBV entry and fusion, and it is unclear whether the authors speculate on the existence of a co-receptor on epithelial cells that engages the tip of D-I domain.

The manuscript is written clearly and properly references the existing literature. My major criticism of the manuscript is that on the basis of available data, it is too premature to exclude the possibility that D-I may contribute to integrin binding and that E1D1 may inhibit integrin binding. The tip of D-I and the KGD motif are not that far from each other. Ideally, the effect of E1D1 and mutations at the tip of D-I on integrin binding should be tested experimentally. At the very least, the authors should state this possibility throughout the manuscript. As another criticism, fusion data should be statistically analyzed and interpreted based on this analysis. Cell surface expression data is also lacking and needs to be provided to allow proper analysis of the fusion data.

Major criticisms:

1. Lines 279-304. Section "Mutations in gL affect membrane fusion with epithelial cells". Firstly, the rationale behind mutagenesis of these residues is not explained. What effect was expected? For example, residue R78 is not described in the prior section as making contacts with E1D1, yet this residue was mutated. Likewise, Y131 was not discussed and is not even shown in Figure 5. So, why was it mutated?

Secondly, cell surface levels of the mutants are not shown. Just because the overall expression levels are similar does not mean the same levels are present on the surface. Showing cell surface expression is necessary for proper evaluation of the fusion results.

Thirdly, mutations had modest effect on fusion, decreasing it by 60% at the most. The number of technical and biological replicates should be listed, and the data should be statistically analyzed before any conclusions regarding significant vs insignificant effect can be made.

Finally, authors should discuss all of the fusion assay results. For example, R78L mutation reduced both B cell and epithelial cell fusion by ~2-fold, yet this effect is not discussed. Some other mutations have comparable, even though, minor effects on both B cell and epithelial cell fusion. What conclusions can be drawn from the location of mutations that significantly affect fusion vs. those that do not? This should be stated clearly.

2. Lines 148-150. In the absence of direct experimental evidence, the conclusion that E1D1 inhibition is not due to competition with integrin binding is premature. Even if E1D1 does not directly block the integrin binding site, its binding could have an allosteric effect on integrin

binding.

3. Lines 303-304 and 328-329. This conclusion that the D-I domain "may form cell-type specific interactions that enhance membrane fusion with epithelial cells" is vague and somewhat premature. Besides certain integrins, no other molecules on the surface of the epithelial cells have been proposed to be involved in EBV entry and fusion, and it is unclear whether the authors refer to the existence of a co-receptor on epithelial cells. More importantly, the possibility that D-I contributes to integrin binding and E1D1 inhibits integrin binding cannot be excluded on the basis of available data.

Other criticisms:

1. Lines 102-103. Lack of an effect of E1D1 on HSV mediated fusion is not shown and is not an important control, so it does not need to be mentioned.

2. Lines 159-162. Authors mention previous studies that identified residues within gp42 N terminus that were critical for binding to gH/gL but do not show them in the structure or discuss how the structure explains their importance for gp42 binding to gH/gL. This information should be added.

3. Lines 222-225 and Figure 4. Authors specifically mention that only 3 out of 10 hydrophobic residues within the hydrophobic groove in gp42 directly contact gH; yet, Figure 4b highlights 5 out of 10 residues. Also, Figure 4a needs to follow the same color scheme as Figure 4b, for clarity.

4. Lines 229-232. The description of the interactions would benefit from a figure showing gH/gL as a surface and gp42 as a cartoon.

5. Lines 235-236. The crystal structure is described as aligning more closely with the closed rather than with the open conformational state of the gH/gL/gp42/HLA complex, which were observed by EM. This conclusion should be illustrated by a side-by-side comparison of the two states and the crystal structure.

6. Throughout the manuscript, please, make it easier for the reader to tell which molecule described residues belong to by explicitly listing the molecule, e.g., gH:H205 or H205gH.

7. Lines 238-243. This speculative section belongs to the Discussion.

8. Lines 250-252. The N terminus of gp42 could block integrin binding by acting allosterically.

For example, it could restrict flexibility of across gH DII-D-IV domains. Ideally, this should be tested experimentally. At the very least, this possibility should be discussed.

9. Supplemental figures 3 and 4 are not referred to anywhere in the main text.

10. Lines 261-262. The structure of D-I is known, so it should be possible to propose how mutations of these residues could disrupt its proper folding.

11. Line 438. Were all three search models were used in separate molecular replacement trials or used in the same trial? This should be clarified.

12. Line 447. Why was occupancy refined at 3.1 angstrom resolution?

13. Figure 5. Panels a and b appear to be related by a 180 degree rotation, but this should be shown on the figure itself. Figure 5 contains a large amount of information and is difficult to follow with the currently used colors.

14. Supplementary figure 1. This gel is smeary and some of the band sizes are confusing. For example, gL in the gH/gL/gp42 sample is running faster than gL in the other 3 samples containing gL. Gp42 in both mixed samples is migrating faster and is labeled gp42 CTLD, but no explanation is provided. Finally, the amount of gp42 in the rightmost lane is very low. Again, this is not explained. Many samples look degraded raising concerns about the quality of protein preps.

15. Supplementary figure 4. The cartoon model does not appear to fit well into the shown density. Authors should show a closeup view of the density and the fit atomic model.

16. Table 1. 2.5% Ramachandran outliers is too high, which raises concerns regarding the quality of the structure. Has the structure been deposited to RCSB and validated? What does CC* represent?

Reviewers' comments and response:

Reviewer #1

And as a non-crystallographer I would like to see fewer structures, a recognition that many are repeated, and most importantly to bring the conclusions out of the supplement and directly into the body of the paper.

We acknowledge that the manuscript has a significant number of structural figures and have deleted Figure 1c as requested by the reviewer. We also removed Figure 4d and moved two panels from Figure 5 to Supplementary Fig. S7. We moved a revised version of Supplementary Fig. S6 into the main Figures, as suggested by the reviewer, to highlight the major conclusions. We have overall 4 out of 7 structural figures in the manuscript and we feel that these 4 figures are necessary for the complete presentation of the results, which focuses on the overview of the structure and 3 key interfaces: (1) between the gp42 N-terminal domain and gH; (2) between the gp42 C-terminal domain and the gH 'KGD' region; and (3) between E1D1 and gL. We feel that each panel of these figures are significant in clarifying the observations to the reader.

The authors need to put all of these new concepts in one place, i.e., a table which summarizes the findings. Now they are spread though-out the manuscript.

We agree with the reviewer's comment and have significantly revised the Discussion section in response, highlighting the observations made by this reviewer in the first paragraph. We are not sure how to incorporate this information into a table, although this might be feasible as part of a review format.

gp42 peptides that block membrane fusion with epithelial cells do NOT physically block the KGD motif, as was previously assumed. The authors suggest that these gp42 peptides indirectly affect gHgL-integrin binding. Has it actually been shown that KGD is where integrin binds to gH? Or is it just assumed?

It has not been formally shown that the 'KGD' motif in gHgL is where integrin binds, but this would be most consistent with the integrin specificity. For this reason, the 'KGD' is referred to only as a 'putative' integrin binding motif for gHgL.

In an earlier paper the authors mutated EBV gH KGD to AAA but they only tested gp42 binding, not integrin binding. Was this tested here and if not why?

We have not been able to reproduce the binding of the gHgL to soluble or cell-bound integrin despite the anticipated high affinity interaction and therefore we have not been able to test this interaction directly. We understand that our results contrast with previously published data and we have been working to understand why this might be.

The gH C-terminal "flap" motif is covered when gp42 binds. Does this change the authors' thinking in the possible function of the gH flap? It had been suggested that this region opens to reveal hydrophobic residues, that then interact with the membrane to promote fusion. If gp42 is covering the flap, it can't open.

The reviewer raises an interesting question regarding the C-terminal "flap" region and its role in membrane fusion. Although it has been suggested that this "flap" region could undergo a conformational change to promote fusion, no conformational changes in this gH domain have been observed. Nonetheless, mutations within gH D-IV do affect membrane fusion and the anchoring of the gp42 N-terminal domain in this region could thereby impact membrane fusion activity. We have added a sentence to that effect in the overall description of the complex in lines 192-195.

Implications...Does your crystal structure (gHgL/gp42/E1D1) differ from either previously published structures (gHgL alone or gp42 alone)? E1D1 has been previously shown to alter gHgL conformation, so are there any significant changes? This needs to be addressed.

We have revised the text on page 7-8 and included a new figure (Supplementary Fig. S6) to address these comments. The root mean square deviation or r.m.s.d from aligning the different structure models are 1.5 Å for gHgL and 0.7 Å for gp42, indicating no major conformational changes in either protein. Earlier studies by us indicated that the gp42 hydrophobic pocket (HP) widens slightly upon HLA binding. The gp42 in our current model is also in the widened state although HLA is absent. Additionally, although some variation in the aligned gH structures in domains D-II to D-IV is observable, the free gHgL structure is of moderate resolution (3.6 Å) and this limits the interpretability of these changes.

"Closed" vs "open" conformational state of gHgL/gp42. The authors claim this crystal is of closed state (line 235). Please remind the reader what this means... does "closed" or "open" align with a gHgL function?

We have revised the text to clarify the 'closed' and 'open' states as suggested and moved this text to the revised discussion. We have also added a panel in Figure 7 (Fig. 7b) highlighting the 'closed' and 'open' states observed earlier by single particle EM in response to the reviewer.

Clearly, all of the action presumably occurs on one side of gH/gL but what about the other side of the complex? Are there changes that take place in structure when gp42 is bound?

As indicated above, we have included a new figure comparing the gHgL structures, which do show differences. However, given the low resolution of the initial gHgL structure we are hesitant to draw significant conclusions from these differences.

Supplemental Fig. 6 is too important to be in the supplement. It needs to be part of the main paper. It summarizes many of the authors' findings.

Supplemental Figure 6 has been moved to the main paper as Figure 7 following the reviewer's suggestion.

But I would like to see more correlations between this structure, the EM structure and the predictions they made previously. And these should be in the Discussion, not in the Results section.

We have significantly revised the text in both the results and discussion sections to draw a number of correlations between the current structure and the previous EM complexes that are included in the new Figure 7 (original Supplemental Figure 6).

Most figure legends are very wordy and repeat the information in the text. Because of this, rather than being informative they become tedious.

The figure legends have been revised throughout to make them more succinct and less redundant with the text.

Fig1b, c: what does the x-axis mean? Is that total amount? ug/ml? ug/ul?

The x-axis in Figures 1b and c has been changed and now refers to the final concentration ($\mu\text{g/ml}$) of E1D1 mAb or Fab added to the cell-cell fusion assay.

Why not show parts of Suppl Fig2 in Fig1?

We have conducted these SPR experiments to examine the binding of E1D1 to gHgL and its complexes with gp42 N-terminal domain and ectodomain. Since the data do not show any significant influence of gp42 on E1D1 binding we feel that this figure does not add significantly to the main points in the text and the binding data is succinctly included in Table 1. Adding the Supplementary Figure to Figure 1 would also double the number of panels in Figure 1 and detract from the points made in panels a-c.

Fig 2C is redundant. Information conveyed could be shown in Fig 2B.

Figure 2 has been revised with panel 2c deleted.

Description for Fig3 is hard to follow, full of details and too long

We have revised the Figure 3 legend to reduce the details and make this easier to follow.

Fig6 gave me the most trouble, may be because this was the only section that I could actually follow. The legend is wordy and confusing.

We have revised Figure 6 and the legend to improve the clarity.

What is the point of it? So, mutations in the E1D1-gHgL interface were made and these mutants have decreased binding to the Ab. The assumption is that these mutants will also have an effect in function. Why? The statement that "the data demonstrate cell-specific modulation" is a bit strong. The effect of these mutations is modest. The data are supposed to add functional weight to the structure but fail to do that.

We have revised the manuscript text on page 13 to better explain the rationale for these mutations.

Because the E1D1 epitope was unexpectedly distant from the putative integrin receptor-binding site, we hypothesized that E1D1's partial inhibition of epithelial cell fusion might be due to an unanticipated functional role of this D-I region. We therefore mutated residues in D-I, not only to examine E1D1 binding, but to investigate whether these impacted epithelial cell fusion selectively. In fact, most of the D-I tip mutants do mirror E1D1 in having more pronounced impact on epithelial cell fusion as compared to B cell fusion. In particular, mutations of the carbohydrate attachment site at gL:N69 significantly increase epithelial cell but not B cell fusion up to 2.5 fold, consistent with our hypothesis. We have revised Figure 6 and the accompanying text to highlight these observations more clearly.

I don't understand what Fig6A is: ELISA? SPR? It looks like ELISA, but it is graphed in such an odd way that I do not know if the negative bars mean no binding or just decreased binding.

Fig. 6a (now Fig. 6b) is CELISA data monitoring the expression of gHgL with a panel of antibodies. The data is presented as the percent change in antibody binding for each mutant vs. wt – this highlights the very selective loss of E1D1 binding by 4 of the mutants that have no impact on other gHgL epitopes monitored by three monoclonals and 1 polyclonal serum (HL800). This presentation demonstrates that the mutations do not affect gHgL expression or folding and makes it easier to observe the few mutations that only affect E1D1 binding, but not the other antibodies. In the percent change in antibody binding graph representation, bars closer to 0 denote wildtype-like binding and bars closer to -100 represent decreased binding, with -100% indicating complete loss of antibody binding.

The assumption that by western all mutants have wt expression is only partially correct. As they are going to assess them for function, I would have liked to see surface levels by ELISA in both epithelial and Daudi cells. What was the Ab used for detection?

As described above, we measured the surface expression of the mutants using a panel of anti-gHgL antibodies and using an anti-HA tag on the wt and mutant gL. We have revised Figure 6 to show the surface expression using the HA tag along with the fusion data so that this can be compared easily. The gHgL surface levels are also monitored by multiple anti-gHgL antibodies (CL40, E1D1, CL59 and HL800) and this data is shown in the revised Figure 6b. Western blot data for all gL mutants has been moved to Supplementary Fig. S8.

What is the control in fusion? It is not indicated in the legend, text or methods.

The control in the fusion assays is gH vector alone without gL. The figure legend has been updated to clarify this point.

The description for Fig6c first covers residues 74-131 and then the glycosylation sites. Therefore, the graphs should be reversed as well.

We have revised the Figure taking into account this comment.

This figure feels like a filler, to have some functional data that was inspired by the structure.

We hope that after clarifying our rationale for carrying out the mutagenesis study, it is clearer to the reviewer that these studies were focused on understanding the mechanism of E1D1 inhibition.

I could not find any reference in the text for Suppl Fig 3 and 4.

In the original manuscript, references to Supplementary Fig. S3 and S4 were made in the Methods section. We have now included references in the main text as well.

Nevertheless, Suppl Fig 4a is a repeat of Fig 2a.

Supplementary Fig. S4a shows electron density used to build the gp42 N-terminal domain, which may be of interest to more technical readers. This information is not shown in Figure 2.

I found Suppl Fig6 more informative than some of the structural descriptions. As it is described and discussed at length, it should be part of the main set of figures.

Supplementary Fig. S6 has been moved to the main text following the reviewer's suggestion.

Reviewer #2 (Remarks to the Author):

Besides certain integrins, no other molecules on the surface of the epithelial cells have been proposed to be involved in EBV entry and fusion, and it is unclear whether the authors speculate on the existence of a co-receptor on epithelial cells that engages the tip of D-I domain.

We have revised the discussion section to address this point. In the revised text we clarify that the mutations in the D-I domain point to an interaction that can enhance membrane fusion with epithelial cells, but these data do not establish whether this is due to an interaction with a co-receptor.

My major criticism of the manuscript is that on the basis of available data, it is too premature to exclude the possibility that D-I may contribute to integrin binding and that E1D1 may inhibit integrin binding. The tip of D-I and the KGD motif are not that far from each other. Ideally, the effect of E1D1 and mutations at the tip of D-I on integrin binding should be tested experimentally. At the very least, the authors should state this possibility throughout the manuscript.

We agree with the reviewer and have revised the text to clarify that we cannot exclude a direct effect of E1D1 on inhibiting integrin binding (e.g. lines 329-330 and 382).

As another criticism, fusion data should be statistically analyzed and interpreted based on this analysis. Cell surface expression data is also lacking and needs to be provided to allow proper analysis of the fusion data.

Cell surface expression data was included in the original figure, but this was not clear to either reviewer. We have modified Figure 6 to better present the fusion and expression data and revised the text and legend to better explain the figures. We have also included statistical analysis of the data.

Lines 279-304. Section "Mutations in gL affect membrane fusion with epithelial cells". Firstly, the rationale behind mutagenesis of these residues is not explained. What effect was expected? For example, residue R78 is not described in the prior section as making contacts with E1D1, yet this residue was mutated. Likewise, Y131 was not discussed and is not even shown in Figure 5. So, why was it mutated?

As described above in response to a similar comment from reviewer #1, we have revised the text to better explain our rationale for the mutagenesis experiments. We hypothesized that E1D1's partial inhibition of epithelial cell fusion might be due to an unanticipated functional role of this D-I region and therefore also included mutations of residues proximal to the E1D1 epitope. This is better explained in the revision.

Secondly, cell surface levels of the mutants are not shown. Just because the overall expression levels are similar does not mean the same levels are present on the surface. Showing cell surface expression is necessary for proper evaluation of the fusion results.

Cell surface expression data in Figure 6, monitored using an anti-HA tag on gL, is now plotted along side the fusion activity and does not vary significantly from wt for the mutants. Panel 6b also shows the surface expression measured using anti-gHgL antibodies.

Thirdly, mutations had modest effect on fusion, decreasing it by 60% at the most. The number of technical and biological replicates should be listed, and the data should be statistically analyzed before any conclusions regarding significant vs insignificant effect can be made.

The experiments were carried out in triplicate (biological replicates) [Lines 554-555]. Statistical tests to indicate significance are updated in the figure (Fig. 6a). We note that the decrease in fusion of ~60% closely mirrors the inhibition observed with E1D1.

Finally, authors should discuss all of the fusion assay results. For example, R78L mutation reduced both B cell and epithelial cell fusion by ~2-fold, yet this effect is not discussed. Some other mutations have comparable, even though, minor effects on both B cell and epithelial cell fusion. What conclusions can be drawn from the location of mutations that significantly affect fusion vs. those that do not? This should be stated clearly.

We have revised the text to more thoroughly discuss the mutants. Lines 311-315 address the effects of R78 mutants. Overall, the mutations show more selective effects on fusion with epithelial cells, which is only partial and consistent with the partial and cell-specific inhibition of E1D1. These data indicate that this region is not absolutely required for membrane fusion, but enhances gHgL activity in epithelial cell fusion.

Lines 148-150. In the absence of direct experimental evidence, the conclusion that E1D1 inhibition is not due to competition with integrin binding is premature. Even if E1D1 does not directly block the integrin binding site, its binding could have an allosteric effect on integrin binding.

We have revised the text to indicate that we cannot rule out allosteric effects on integrin binding as suggested.

Lines 303-304 and 328-329. This conclusion that the D-I domain "may form cell-type specific interactions that enhance membrane fusion with epithelial cells" is vague and somewhat premature. Besides certain integrins, no other molecules on the surface of the epithelial cells have been proposed to be involved in EBV entry and fusion, and it is unclear whether the authors refer to the existence of a co-receptor on epithelial cells. More importantly, the possibility that D-I contributes to integrin binding and E1D1 inhibits integrin binding cannot be excluded on the basis of available data.

We have revised the text to indicate that this region of D-I appears to influence the efficiency of epithelial cell fusion more than B cell fusion, based on the E1D1 inhibition and mutagenesis results, but refrain from extrapolating this to hypothetical cell-type specific interactions.

Lines 102-103. Lack of an effect of E1D1 on HSV mediated fusion is not shown and is not an important control, so it does not need to be mentioned.

The text has been revised as suggested.

Lines 159-162. Authors mention previous studies that identified residues within gp42 N terminus that were critical for binding to gH/gL but do not show them in the structure or discuss how the structure explains their importance for gp42 binding to gH/gL. This information should be added.

The residues identified from our previous study, and those which make direct interactions in our structure are now highlighted in Fig. 3a in bold red.

Lines 222-225 and Figure 4. Authors specifically mention that only 3 out of 10 hydrophobic residues within the hydrophobic groove in gp42 directly contact gH; yet, Figure 4b highlights 5 out of 10 residues. Also, Figure 4a needs to follow the same color scheme as Figure 4b, for clarity.

Fig. 4 a and b have been revised as suggested.

Lines 229-232. The description of the interactions would benefit from a figure showing gH/gL as a surface and gp42 as a cartoon.

We made this figure as suggested by the reviewer but did not think that this representation of the interactions added significantly to our current figures. In addition, given Reviewer #1's suggestions to reduce the amount of structural information presented, we felt that it was best not to include this representation.

Lines 235-236. The crystal structure is described as aligning more closely with the closed rather than with the open conformational state of the gH/gL/gp42/HLA complex, which were observed by EM. This conclusion should be illustrated by a side-by-side comparison of the two states and the crystal structure.

We have added Figure 7 as a revision of our previous Supplementary Fig. S6 to show these comparisons more clearly.

Throughout the manuscript, please, make it easier for the reader to tell which molecule described residues belong to by explicitly listing the molecule, e.g., gH:H205 or H205gH.

We have revised the text to label residues by molecule using the gH:H205 format for all molecules as suggested.

Lines 238-243. This speculative section belongs to the Discussion.

We have revised the results section to simply point out the two histidines and moved these lines to the discussion as suggested.

Lines 250-252. The N terminus of gp42 could block integrin binding by acting allosterically. For example, it could restrict flexibility of across gH DII-D-IV domains. Ideally, this should be tested experimentally. At the very least, this possibility should be discussed.

We have revised the text (lines 273-274) to more clearly point out this possibility. As mentioned above, we have had encountered difficulties in observing the integrin:gHgL interaction to test this experimentally.

Supplemental figures 3 and 4 are not referred to anywhere in the main text.

We have added references to these figures in the main text as they were previously only referred to in the Methods sections.

Lines 261-262. The structure of D-I is known, so it should be possible to propose how mutations of these residues could disrupt its proper folding.

We have revised the text to clarify this point (lines 276-277). L65 and L69 are in the hydrophobic core of D-I near the 'D-I/D-II Linker helix', consistent with the mutations destabilizing the folding of the D-I domain and gL.

Line 438. Were all three search models were used in separate molecular replacement trials or used in the same trial? This should be clarified.

All three search models were used in the same phaser run for each Fab elbow angle and this information is included in the revised text.

Line 447. Why was occupancy refined at 3.1 angstrom resolution?

The electron density maps for gH:H426 were clearly consistent with two different side chain rotamer conformations. Although it is unusual for a 3.1 Å model to have alternate conformations modeled, we felt that the this was appropriate given the observations. The occupancies of just these two single side chain conformers were refined.

Figure 5. Panels a and b appear to be related by a 180 degree rotation, but this should be shown on the figure itself. Figure 5 contains a large amount of information and is difficult to follow with the currently used colors.

Figure 5 panels a and b are indeed related by a 180 degree rotation along the y direction. This has been clarified in the revised Figure 5. We have also modified the labeling and coloring in the figure to make it easier to follow.

Supplementary figure 1. This gel is smeary and some of the band sizes are confusing. For example, gL in the gH/gL/gp42 sample is running faster than gL in the other 3 samples containing gL. Gp42 in both mixed samples is migrating faster and is labeled gp42 CTLD, but no explanation is provided. Finally, the amount of gp42 in the rightmost lane is very low. Again, this is not explained. Many samples look degraded raising concerns about the quality of protein preps.

We have revised this figure to provide a clearer example of our protein preparations.

Supplementary figure 4. The cartoon model does not appear to fit well into the shown density. Authors should show a closeup view of the density and the fit atomic model.

We have revised the figure to more clearly show the electron density and included a close up view as requested.

Table 1. 2.5% Ramachandran outliers is too high, which raises concerns regarding the quality of the structure. Has the structure been deposited to RCSB and validated? What does CC represent?*

We have deposited the structure in the RCSB and validated the model. Updated statistics on the refined structure are shown in Supplementary Table 1, showing fewer Ramachandran outliers. CC* is a data quality statistic that represents an estimate of the cross correlation between the observed dataset and the "true" (noise-free) signal as described by Karplus and Diederichs (Science 2012 May 25; 336(6084):1030–33). CC* provides a better tool for estimating the appropriate cutoff resolution for weak data.

Reviewers' Comments:

Reviewer #1 (Remarks to the Author):

The authors have satisfied all of the criticisms of Reviewer 1. Great job...

Reviewer #2 (Remarks to the Author):

Authors have satisfactorily addressed all of my criticisms.